# Latent Diffusion Pretraining for Crystal Property Prediction

## Abstract

Fast and accurate prediction of crystal properties is a central challenge in new materials design. Graph Neural Networks have emerged as powerful tools for this task due to their ability to encode the local structural environment of atoms within a crystal. However, these models are data hungry and in practice labeled data for crystal properties are very scarce. Pretrain–finetuning strategies, particularly those based on diffusion models, have shown promise in addressing these limitations. In this work, we introduce a novel latent-diffusion based pretraining framework designed to mitigate the data scarcity issue. Our approach integrates a Variational Autoencoder (VAE) with a diffusion model during the pretraining stage. The VAE encoder maps 3D crystal structures into a smooth latent space, within which the diffusion process is applied. This latent diffusion pretraining enables the graph encoder to effectively capture structural and chemical semantics from large scale unlabeled data, which can then be finetuned for specific property prediction tasks. Comprehensive experiments on popular DFT datasets for property prediction reveal that CrysLDNet significantly outperforms both training-from-scratch and pretrained baselines, with average improvements of *6.93%* and *7.83%* on JARVIS and MP over the second-best baseline. Additionally, the learned representations remain robust under sparse data conditions and are expressive enough to correct DFT errors when finetuned with limited experimental data.

## 1 Introduction

Crystal materials drive advancements in energy, electronics, healthcare, transportation, and infrastructure (Butler et al., 2018; Desiraju, 2002). A key step in the materials design pipeline is fast and accurate prediction of chemical properties for newly discovered crystals. Over the years, Density Functional Theory (DFT) (Orio et al., 2009) has been widely used to estimate various chemical properties; however, its high computational cost makes the screening process inefficient. With the rise of machine learning, data-driven approaches (Gaultois et al., 2016; Lu et al., 2018; Gómez-Bombarelli et al., 2016; Xue et al., 2016) have emerged as powerful alternatives, delivering crystal property predictions with accuracy comparable to DFT at a fraction of the computational cost. In particular, graph neural network (GNN) based models (Xie & Grossman, 2018; Chen et al., 2019; Louis et al., 2020; Park & Wolverton, 2020; Schmidt et al., 2021; Choudhary & DeCost, 2021) have gained particular prominence, where they represent 3D material structures as multi-edge graphs and apply GNNs to learn structural representations optimized for downstream property prediction.

Like other deep neural networks, these models require large labeled datasets to train its parameters for accurate predictions. In crystal, however, labeled data are scarce and vary widely across properties, while curating such datasets through simulations or experiments is highly resource and time intensive. In contrast, vast amounts of unlabeled crystal data with only 3D structural information are readily available. Recent studies have explored self-supervised pretraining on large collections of unlabeled 3D crystal structures, enabling GNN encoders to learn underlying chemical and structural semantics and produce meaningful embeddings for downstream property prediction. For instance, CrysXPP (Das et al., 2022) introduced unsupervised pretraining followed by fine-tuning on property-labeled data, while CrysGNN (Das et al., 2023b) extended this idea with large-scale self-supervised pretraining. Similarly, Crystal Twins (Magar et al., 2022) uses self supervised learning for pre-training CGCNN encoder by using barlow twins loss function. However, these approaches rely on CGCNN as the encoder, which limits their overall expressivity.

Lately, CrysDiff (Song et al., 2024) and DPF (Shen et al., 2025a) have explored diffusion-based pretraining frameworks for crystal structure reconstruction task. CrysDiff employs a joint denoising diffusion model to reconstruct crystal structures from atomic compositions during pretraining, and

during finetuning it conditions diffusion on target property values with fixed structures. In contrast, DPF perturbs atom types, positions, and lattice constants during pretraining, then reconstructs the native crystal structure, and the learned representation further finetuned for downstream property prediction. A key limitation of these diffusion-based pretraining methods is that they operate on high-dimensional feature spaces, where they jointly model atom types, fractional coordinates, and lattice structures. Fractional coordinates, following a wrapped normal distribution, are usually modeled with score-based approaches (Song et al., 2020); atom types, being categorical, with discrete diffusion models like D3PM (Austin et al., 2021); and continuous lattice structures with DDPMs (Ho et al., 2020). This heterogeneous modeling demands complex denoising architectures and many diffusion steps to achieve high-quality crystal representations. Moreover, since these methods operate directly in the input feature space, which is inherently non-smooth, the resulting representations tend to be less expressive and lead to suboptimal performance in property prediction tasks.

In this work, we adopt a latent diffusion-based pretraining framework to overcome the limitations of conventional diffusion-based pretraining. Crystal properties such as formation energy, band gap, and others are fundamentally determined by the atomic arrangement and overall structure. Therefore, learning enriched and expressive representations that can reconstruct both atomic composition and 3D structure is more meaningful and beneficial for downstream property prediction tasks. Building on this insight, we propose CrysLDNet, a novel pretraining strategy based on latent diffusion that learns robust and expressive crystal representations to improve downstream property prediction. Our methodology consists of two stages: pretraining on large-scale, unlabeled crystal datasets and fine-tuning on smaller property-labeled datasets. The pretraining framework consists of two key modules: a Variational Autoencoder (VAE) and a Latent Diffusion Model (LDM). The VAE encoder compresses high-dimensional crystal structures into a compact latent space, while the decoder reconstructs the original structures from these latent embeddings. The LDM then operates on this latent space, generated by the encoder, by progressively adding noise to the latent representations and learning to denoise them via a transformer-based module. Through this two-step pretraining, the encoder is guided to capture the structural semantics of 3D crystal materials, generating enriched latent representations well-suited for property prediction. We present empirical evidence (Section 5.2.1, Fig. 3) showing that crystal embeddings learned through latent diffusion–based pretraining can more effectively reconstruct both atomic composition and 3D structure compared to conventional pretraining methods that apply diffusion directly in the feature space. Finally, these pretrained representations are fine-tuned with a property predictor using limited labeled data, significantly enhancing both efficiency and accuracy in downstream tasks.

To sum up, our novel contributions in this work are as follows:

- To the best of our knowledge, this work is the first to explore a latent diffusion–based pretraining framework for crystal property prediction.
- We propose CrysLDNet[1], a latent diffusion–based pretraining framework that learns robust and enriched crystal representations via VAE encoding and latent diffusion denoising, enabling efficient and accurate downstream property prediction.
- Extensive experiments on widely used DFT datasets for benchmark property prediction tasks show that CrysLDNet outperforms both training-from-scratch and pretrained baselines by a good margin.
- Moreover, our results demonstrate that the learned representations are robust in sparse data regimes and sufficiently expressive to mitigate DFT error bias when finetuned with limited experimental data.

## 2 PRELIMINARIES

### 2.1 CRYSTAL REPRESENTATION

Crystal materials can be viewed as a 3D point cloud of atoms arranged in an orderly repeating pattern. They are modeled by a minimal unit cell, which contains all constituent atoms at specific coordinates. This unit cell repeats itself infinitely in three-dimensional space along a regular lattice and forms periodic structures. For a material with $N$ atoms in its unit cell, the structure can be defined as $\boldsymbol{M} = (\boldsymbol{A}, \boldsymbol{X}, \boldsymbol{L})$. *Atom Type Matrix:* $\boldsymbol{A} = [\mathbf{a_1}, \mathbf{a_2}, ..., \mathbf{a_N}]^T \in \mathbb{R}^{N \times k}$, where each $\mathbf{a_i}$ is a one-hot vector denoting the atomic type of the $i^{th}$ atom, and $k$ is the maximum number of possible atom types. *Coordinate Matrix:* $\boldsymbol{X} = [\mathbf{x_1}, \mathbf{x_2}, ..., \mathbf{x_N}]^T \in \mathbb{R}^{N \times 3}$, where $\mathbf{x_i} \in \mathbb{R}^3$ represents the 3D coordinates of the $i^{th}$ atom in the unit cell. *Lattice Matrix:* $\boldsymbol{L} = [\mathbf{l_1}, \mathbf{l_2}, \mathbf{l_3}]^T \in \mathbb{R}^{3 \times 3}$,

---

[1]Source code is provided in the Supplementary Material.

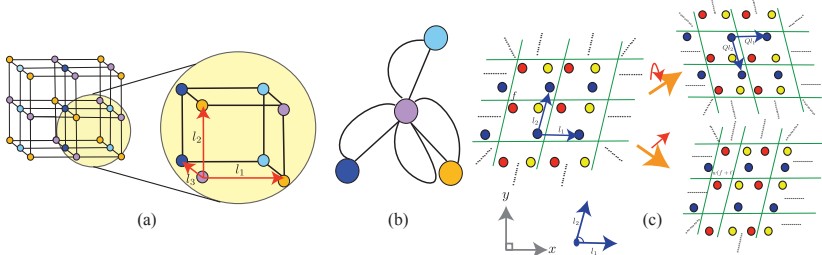

Figure 1: (a) A periodic crystal structure represented as a point cloud of atoms arranged in repeating patterns, along with a magnified view of a unit cell. (b) A multigraph representation of the unit cell. (c) Rotational, translational, and periodic symmetries of the crystal.

which specifies how the unit cell repeats itself in 3D space along directions $l_1, l_2$, and $l_3$ to form the periodic crystal. Formally, its infinite periodic structure can be represented as: $\hat{X} = \{\hat{\mathbf{x}}_i | \hat{\mathbf{x}}_i = \mathbf{x}_i + \sum_{j=1}^{3} k_j l_j\}$ and $\hat{A} = \{\hat{\mathbf{a}}_i | \hat{\mathbf{a}}_i = \mathbf{a_i}\}$ where $k_1, k_2, k_3, i \in Z, 1 \le i \le N$.

**Symmetry in Crystal Structure:.** Crystal materials exhibit fundamental physical symmetries that any learned representation must respect. One such property is rotational invariance, which ensures that rotating the atom coordinates and lattice matrices with any orthogonal matrix $\mathbf{Q}$ results in an equivalent representation of the same material. Another key property is periodic translation invariance, meaning that translating the atom coordinates by any arbitrary vector and applying periodic wrapping does not alter the crystal structure. Also, since atoms in the unit cell repeat infinitely along the lattice vectors, multiple choices of unit cells and coordinate matrices can equivalently represent the same material. A formal definition of these invariance properties is provided in Appendix A.

## 2.2 MULTI-GRAPH CONSTRUCTION FOR CRYSTALS

Most of the state-of-the-art GNN frameworks for crystal property prediction realize a crystal material as a multi-graph structure $\mathcal{G} = (\mathcal{V}, \mathcal{E}, \mathcal{X}, \mathcal{F})$ as shown in Fig. 1 (b) as proposed by CGCNN Xie & Grossman (2018). $\mathcal{G}$ is an undirected weighted multi-graph where $\mathcal{V}$ denotes the set of nodes or atoms present in a unit cell of the crystal structure. $\mathcal{E} = \{(u, v, k_{uv})\}$ denotes a multi-set of node pairs and $k_{uv}$ denotes number of edges between a node pair $(u, v)$. $\mathcal{X} = \{x_u | u \in \mathcal{V}\}$ denotes 92 dimensional node feature set proposed by CGCNN Xie & Grossman (2018). It includes different chemical properties like electronegativity, valance electron, covalent radius, etc. Finally, $\mathcal{F} = \{\{s^k\}_{(u,v)} | (u, v) \in \mathcal{E}, k \in \{1..k_{uv}\}\}$ denotes the multi-set of edge weights where $s^k$ corresponds to the $k^{th}$ bond length between a node pair $(u, v)$, which signifies the inter-atomic bond distance between two atoms.

## 3 RELATED WORK: CRYSTAL PROPERTY PREDICTION

In recent times, graph neural network(GNN) models become popular tools for crystal property prediction. Earlier approaches Xie & Grossman (2018); Chen et al. (2019); Louis et al. (2020); Park & Wolverton (2020); Schmidt et al. (2021); Choudhary & DeCost (2021); Yan et al. (2022); Lin et al. (2023) construct a multi-edge graph from the 3D crystal structure and apply a GNN model to encode the neighborhood structural information around an atom. Building on this, numerous studies have proposed various GNN variants. ALIGNN Choudhary & DeCost (2021) incorporates bond angular information among edges to capture many-body interactions; whereas Matformer Yan et al. (2022) is designed to be invariant to periodicity, enabling it to explicitly capture repeating patterns. Data scarcity remains a significant challenge in this field, motivating the development of various graph pretraining strategies. CrysXPP (Das et al., 2022) and CrysGNN (Das et al., 2023b) leveraged unsupervised pretraining followed by fine-tuning and knowledge distillation on property-labeled data. Similarly, Crystal Twins (Magar et al., 2022) applied self-supervised learning to pretrain the CGCNN encoder using the Barlow Twins loss. More recently, diffusion-based pretraining frameworks have been explored for crystal structure reconstruction. CrysDiff (Song et al., 2024) employs a joint denoising diffusion model to reconstruct crystal structures from atomic compositions during pretraining, and during finetuning it conditions diffusion on target property values while keeping structures fixed. In contrast, DPF (Shen et al., 2025a) perturbs atom types, positions, and lattice

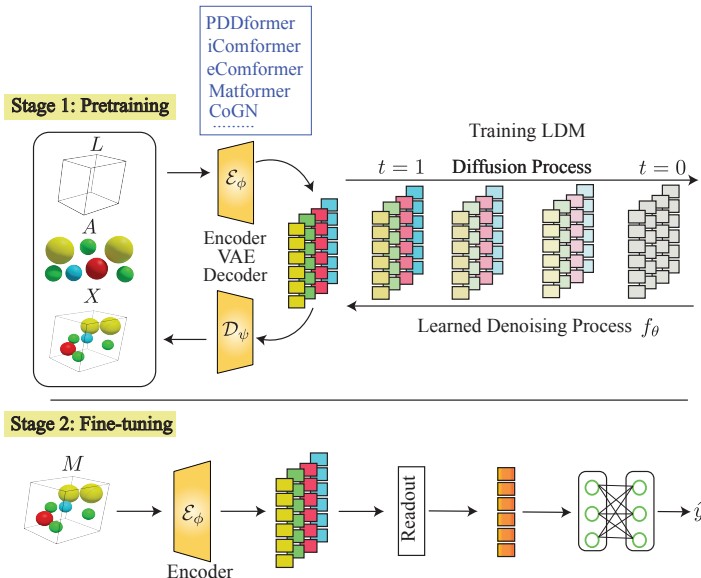

Figure 2: Overview of our proposed pretrain–finetune framework, CrysLDNet. In the pretraining stage, a VAE encodes crystal structures into latent representations, on which the LDM is applied to refine the encoded latent space. CrysLDNet is designed to be fully agnostic to the choice of different backbone encoder architectures. The pretrained encoder is then finetuned on property-labeled data for downstream tasks.

constants during pretraining, reconstructs the native crystal structure, and finetunes the learned representation for downstream property prediction. We provide a comprehensive review of related literature in Appendix B

## 4 PROPOSED METHODOLOGY : CRYSLDNET

### 4.1 PROBLEM STATEMENT

The goal of the crystal property prediction task is, given any material $\mathcal{M} = (A, X, L)$, to predict a downstream target property value $y$. Our proposed framework, CrysLDNet, addresses this problem through latent diffusion–based pretraining followed by finetuning. Specifically, we first leverage all available *unlabeled crystal data* $\mathcal{D}_u = \{\mathcal{M}\}_i$ to pretrain CrysLDNet, enabling it to capture intrinsic structural and chemical patterns of crystal graphs. Subsequently, we utilize a training set of *property-tagged crystal data* $\mathcal{D}_p = (\mathcal{M}_i, y_i)$ to finetune the model on the target property prediction task. During finetuning the pretrained representations are further refined and optimized for the specific downstream property.

### 4.2 CRYSLDNET PRETRAINING

The objective of the pretraining stage is to enable the model to effectively learn and capture the structural and chemical characteristics of crystal materials from a large corpus of unlabeled data. To achieve this, we introduce a latent space diffusion pretraining strategy. The framework consists of two core components: a Variational Autoencoder, producing the latent representation of the crystal 3D structure and a Diffusion Model (DM), operating in a smoother, lower-dimensional latent space.

#### 4.2.1 VARIATIONAL AUTOENCODER (VAE)

Our first objective is to encode the 3D crystal geometry into a lower-dimensional latent space that is both meaningful and preserves the physical symmetries inherent to crystal structures. To accomplish this, we employ a Variational Autoencoder (VAE) framework consisting of an encoder $\mathcal{E}_\phi$ and a decoder $\mathcal{D}_\psi$. The encoder module takes the crystal material $M = (A, X, L)$ as input and encodes atomic types, coordinates and lattice structure into the latent space:

$$Z = \mathcal{E}_\phi(A, X, L) \tag{1}$$

---

**Algorithm 1** Pretraining Algorithm of CrysLDNet

---

1: **Input:** Crystal Material $\boldsymbol{M} = (\boldsymbol{A}, \boldsymbol{X}, \boldsymbol{L})$, Property Value $y$, Encoder $\mathcal{E}_\psi$, Decoder $\mathcal{D}_\phi$, and Denoising Network $\mathcal{F}_\theta$.

2: **Stage-1: Training VAE**
3: **repeat**
4:     $\boldsymbol{Z} \leftarrow \mathcal{E}_\phi(\boldsymbol{A}, \boldsymbol{X}, \boldsymbol{L})$
5:     $\tilde{\boldsymbol{A}}, \tilde{\boldsymbol{X}}, \tilde{\boldsymbol{L}} \leftarrow \mathcal{D}_\psi(\boldsymbol{Z})$
6:     $\mathcal{L}^A = \text{CrossEntropyLoss}(\tilde{\boldsymbol{A}}, \boldsymbol{A})$
7:     $\mathcal{L}^L = \|\tilde{\boldsymbol{L}} - \boldsymbol{L}\|_2^2$
8:     $\mathcal{L}^X = \|\tilde{\boldsymbol{X}} - \boldsymbol{X}\|_2^2$
9:     Minimize $\mathcal{L}^A + \mathcal{L}^X + \mathcal{L}^L + \lambda \mathcal{L}_{reg}$
10:    Update parameters of both $\mathcal{E}_\psi$ and $\mathcal{D}_\phi$
11: **until** Converged

12: **Stage-2: Training LDM**
13: **repeat**
14:    $\boldsymbol{Z}^1 \leftarrow \mathcal{E}_\phi(\boldsymbol{A}, \boldsymbol{X}, \boldsymbol{L})$ // Clean Latent
15:    Sample $t \sim \mathcal{U}(0, 1)$
16:    Sample Noise $\boldsymbol{Z}^0 \sim N(0, 1)^{N \times d}$
17:    $\boldsymbol{Z}^t = (1 - t)\, \boldsymbol{Z}^0 + t\, \boldsymbol{Z}^1$
18:    $\bar{\boldsymbol{Z}}^1 = \mathcal{F}_\theta(\boldsymbol{Z}^t, t)$
19:    $\mathcal{L}_{LDM} = \frac{1}{(1-t)^2} \frac{1}{N} \sum_{i=0}^{N} \left\| \boldsymbol{z}_i^1 - \bar{\boldsymbol{z}}_i^1 \right\|^2$
20:    Minimize $\mathcal{L}_{LDM}$ and Update parameters of both $\mathcal{E}_\phi$ and $\mathcal{F}_\theta$
21: **until** Converged

---

where $\boldsymbol{Z} = \{\boldsymbol{z_h}\} \in \mathbb{R}^{N \times d}$, where $N$ is the number of atoms, is the set of node embeddings of the crystal materials. A key criterion in designing this encoder is that the learned latent space must preserve the physical symmetries of the crystal structure and invariance properties. Hence as encoder, we employ the Matformer network Yan et al. (2022), which ensures that the learned representations satisfy invariance to translation, rotation, and periodic transformations. Further, the decoder $\mathcal{D}_\psi$ is trained to reconstruct the 3D crystal structure. In specific, we utilize three separate MLPs: one dedicated to reconstructing atomic types, another for atomic coordinates, and a third for the lattice structure from the latent representations:

$$\tilde{\boldsymbol{A}}, \tilde{\boldsymbol{X}}, \tilde{\boldsymbol{L}} = \mathcal{D}_\psi(\boldsymbol{Z}) \tag{2}$$

The whole network is trained end-to-end using a regularized reconstruction loss:

$$\mathcal{L}_{\text{VAE}} = \mathcal{L}_{\text{recon}}^A + \mathcal{L}_{\text{recon}}^X + \mathcal{L}_{\text{recon}}^L + \alpha \mathcal{L}_{\text{reg}} \tag{3}$$

Here, $\mathcal{L}_{\text{recon}}^A$, $\mathcal{L}_{\text{recon}}^X$ and $\mathcal{L}_{\text{recon}}^L$ represent the reconstruction losses for atom types, coordinates and lattice structure, respectively. By design, we use cross-entropy loss for $\boldsymbol{A}$ and $l_2$ loss for $\boldsymbol{X}$ and $\boldsymbol{L}$. $\mathcal{L}_{\text{reg}} = d_{\text{KL}}\{q_\phi(\boldsymbol{Z}|\boldsymbol{A}, \boldsymbol{X}, \boldsymbol{L}) || p(\boldsymbol{Z})\}$ denotes KL divergence that measures how much the learned latent distribution $q_\phi(\boldsymbol{Z} \mid \boldsymbol{A}, \boldsymbol{X}, \boldsymbol{L})$ differs from the prior distribution $p(\boldsymbol{Z})$ (commonly a standard gaussian distribution). This regularization term constrains the variance of latent embeddings, making them more stable and suitable for learning latent diffusion models.

### 4.2.2 LATENT DIFFUSION MODEL (LDM)

The VAE encoder $\mathcal{E}_\phi$ projects crystal materials into a smoother, lower-dimensional latent space. To further enhance these latent representations, we apply a diffusion model over this space. Following Joshi et al. (2025), we adopt flow matching (Lipman et al., 2022) or gaussian diffusion (both formulations are same (Gao et al., 2024)), where noise is iteratively added to the latent embeddings produced by $\mathcal{E}_\phi$ over $t$ steps to transform them toward a base distribution, and a denoising process is then employed to recover the original latent representations.

We formulate our approach by employing linear interpolation between a standard Gaussian base distribution and the target distribution defined by the VAE encoder's latent representations of 3D crystal structures. Specifically during training, given a material structure $\boldsymbol{M} = (\boldsymbol{A}, \boldsymbol{X}, \boldsymbol{L})$, we first encode it into a lower-dimensional latent representation $\boldsymbol{Z}$ using the VAE encoder $\mathcal{E}_\phi$. For convenience, we denote $\boldsymbol{Z}$ as $\boldsymbol{Z}^1$, representing a clean training sample at time $t = 1$. Next, we sample a noisy latent variable $\boldsymbol{Z}^0$ at time $t = 0$ from a $d$-dimensional standard Gaussian distribution $\mathcal{N}(0, 1)^d$, followed by zero-centering through subtraction of its per-channel mean. Finally, we construct an interpolated noisy sample $\boldsymbol{Z}^t$ at a randomly sampled time step $t \sim \mathcal{U}(0, 1)$ using linear interpolation: $\boldsymbol{Z}^t = (1 - t)\, \boldsymbol{Z}^0 + t\, \boldsymbol{Z}^1$ Therefore, along the trajectory from the noisy latent at step $t$ to the clean latent, we define the ground-truth conditional vector field $u_t(\boldsymbol{Z}^t \mid \boldsymbol{Z}^1)$ as:

$$u_t(\boldsymbol{Z}^t \mid \boldsymbol{Z}^1) = \frac{\boldsymbol{Z}^1 - \boldsymbol{Z}^t}{1 - t} \tag{4}$$

By integrating this vector field $u_t(\mathbf{Z}^t \mid \mathbf{Z}^1)$ over time, latent samples drawn from the noisy Gaussian distribution are transformed into the true latent representations from the target distribution. We train a denoising network $\mathcal{F}_\theta$ to approximate the conditional vector field $u_t(\mathbf{Z}^t \mid \mathbf{Z}^1)$. The network takes as input the intermediate noisy latent $\mathbf{Z}^t$ along with the time step $t$ and predicts the corresponding clean latent representation as: $\bar{\mathbf{Z}}^1 = \mathcal{F}_\theta(\mathbf{Z}^t, t)$. The denoiser is optimized by minimizing the mean squared error (MSE) loss between the predicted conditional vector field and the ground-truth conditional vector field, formulated as:

$$\mathcal{L}_{\text{LDM}} = \frac{1}{N} \sum_{i=0}^{N} \left\| \frac{\mathbf{z}_i^1 - \mathbf{z}_i^t}{1-t} - \frac{\bar{\mathbf{z}}_i^1 - \mathbf{z}_i^t}{1-t} \right\|^2 = \frac{1}{(1-t)^2} \frac{1}{N} \sum_{i=0}^{N} \left\| \mathbf{z}_i^1 - \bar{\mathbf{z}}_i^1 \right\|^2 \tag{5}$$

As $\mathcal{F}_\theta$, we employ the Diffusion Transformer (DiT) (Peebles & Xie, 2023) architecture. During training, both the VAE encoder and the Diffusion Transformer are optimized jointly using the loss in Eq.5. The role of the encoder is to map 3D crystal structures into latent representations, while the Diffusion Transformer is trained to predict the noise given the noisy latent input. Since the VAE encoder (Matformer network) has already been pretrained with the autoencoding loss in Eq.3, it is further refined at this stage, leading to latent representations that are more enriched and expressive, which will enhance property prediction performance.

**Backbone-Agnostic Design.** A key strength of our CrysLDNet pretraining framework is that it is fully agnostic to the choice of backbone encoder architecture. The crystal graph encoder used in the VAE and the downstream property predictor can be replaced with any crystal-GNN, EGNN or transformer architecture without modifying the rest of the pipeline. All other components like the latent diffusion model, decoder, and task-specific heads, remain unchanged regardless of the encoder choice. This modular design enables seamless substitution of existing backbones (e.g., CGCNN, ALIGNN, DimeNet++, CoGN, Equiformer) by simply plugging them into the encoder slot. Looking ahead, any future, more powerful transformer models can be seamlessly integrated into our pretrain–finetune paradigm, and we expect them to improve performance further.

### 4.3 CRYSLDNET FINE-TUNING

During the pretraining phase, first through the VAE and subsequently via the LDM, the encoder $\mathcal{E}_\phi$ progressively captures meaningful chemical and structural semantics. Building on this, we design a property predictor tailored to specific material properties, leveraging the knowledge learned by the encoder. The property predictor is composed of the previously pretrained encoder, followed by several multi-layer perceptron (MLP) layers, and is fine-tuned for downstream property prediction tasks using the limited available labeled dataset. We begin by generating node-level representations using the encoder function, as described in Eq. 1. Next, a symmetric READOUT function is applied to obtain a graph-level representation $\mathbf{Z}_g$, ensuring invariance to node orderings. Finally, this aggregated representation is passed through an MLP, which predicts the desired material property. Formally, the property predictor can be expressed as:

$$\hat{y} = \text{MLP}_\lambda\big(\text{READOUT}\{\mathcal{E}_\phi(\mathcal{M})\}\big), \tag{6}$$

where $\hat{y}$ is the predicted property value. The network is fine-tuned end-to-end using mean square error (MSE) objective function between predicted $\hat{y}$ and true property values $y$:

$$\min_{\phi,\psi} \mathcal{L}_{\text{MSE}} = \|\hat{y} - y\|^2 \tag{7}$$

By leveraging the pretrained encoder, we transfer its rich encoded knowledge into the property predictor, allowing it to benefit directly from the representations learned during the pre-training stage. During fine-tuning the pretrained representations are further refined and optimized for the specific downstream property. This significantly reduces the reliance on large-scale property-labeled datasets, enabling effective property prediction even with limited labeled data.

## 5 EXPERIMENTS

### 5.1 DATASETS FOR PRETRAINING AND DOWNSTREAM TASKS

For pretraining, we follow prior work (Shen et al., 2025a) and use 380,740 crystal structures filtered from the recent GNoME dataset (Merchant et al., 2023). Following (Shen et al., 2025a) we exclude entries that are duplicates of downstream datasets or lack physical or chemical significance. A detailed description of the pre-trained dataset is provided in Table 5 in Appendix. Further, for

| Property | Supervised Models (Train-from-scratch) | | | | | | Pretrain-finetune Models | | | | | |
|---|---|---|---|---|---|---|---|---|---|---|---|---|
| | CGCNN | SchNet | MEGNet | GATGNN | ALIGNN | Matformer | CrysXPP | Crystal Twins | CrysGNN | CrysDiff | DPF | CrysLDNet |
| Formation Energy | 0.063 | 0.045 | 0.047 | 0.047 | 0.033 | _0.033_ | 0.062 | 0.042 | 0.056 | **0.029** | **0.029** | **0.029** |
| Bandgap (OPT) | 0.200 | 0.190 | 0.145 | 0.170 | 0.142 | 0.137 | 0.190 | 0.160 | 0.183 | 0.131 | _0.122_ | **0.120** |
| Total Energy | 0.078 | 0.047 | 0.058 | 0.056 | 0.037 | 0.035 | 0.072 | 0.050 | 0.069 | 0.034 | _0.032_ | **0.029** |
| Ehull | 0.170 | 0.140 | 0.084 | 0.120 | 0.076 | 0.064 | 0.139 | 0.132 | 0.130 | 0.062 | _0.059_ | **0.045** |
| Bandgap (MBJ) | 0.410 | 0.430 | 0.340 | 0.510 | 0.310 | 0.300 | 0.378 | 0.374 | 0.371 | _0.287_ | 0.311 | **0.280** |
| Bulk Modulus (Kv) | 14.47 | 13.25 | 14.20 | 14.32 | 10.40 | 11.21 | 13.61 | 13.41 | 13.42 | _9.875_ | 10.43 | **9.818** |
| Shear Modulus (Gv) | 11.75 | 11.12 | 12.25 | 12.48 | 9.481 | 10.76 | 11.20 | 11.18 | 11.07 | _9.191_ | 9.596 | **9.108** |
| SLME (%) | 8.022 | 7.431 | 7.213 | 7.504 | 5.146 | 5.260 | 5.110 | _4.967_ | 5.452 | 5.030 | 5.129 | **4.636** |
| Spillage | 0.454 | 0.409 | 0.445 | 0.431 | 0.389 | 0.398 | 0.363 | 0.393 | 0.374 | _0.358_ | _0.358_ | **0.349** |
| Formation Energy | 0.031 | 0.033 | 0.030 | 0.033 | 0.022 | 0.021 | 0.034 | 0.034 | 0.033 | – | _0.020_ | **0.019** |
| Bandgap (OPT) | 0.292 | 0.345 | 0.307 | 0.280 | 0.218 | 0.211 | 0.269 | 0.269 | 0.266 | – | _0.203_ | **0.188** |
| Bulk Modulus (Kv) | 0.047 | 0.066 | 0.060 | 0.045 | 0.051 | 0.043 | 0.055 | 0.051 | 0.043 | – | _0.042_ | **0.038** |
| Shear Modulus (Gv) | 0.077 | 0.099 | 0.099 | 0.075 | 0.078 | _0.073_ | 0.084 | 0.082 | 0.076 | – | _0.073_ | **0.066** |

Table 1: Summary of MAE results for various properties on JARVIS-DFT (top) and Materials Project (bottom). For CrysDiff, due to the unavailability of its code and the absence of reported results on the MP dataset in the original paper, we denote these entries with "–". The best and second-best performances are shown in bold and underlined, respectively.

| Property | Supervised Models (Train-from-scratch) | | | | | | | | Pretrain-finetune Models | | | |
|---|---|---|---|---|---|---|---|---|---|---|---|---|
| | CoGN | DimeNet++ | Equiformer | Matformer | PotNet | eComformer | iComformer | PDDformer | CrysLDNet (Matformer) | CrysLDNet (eComformer) | CrysLDNet (iComformer) | CrysLDNet (PDDformer) |
| Formation Energy | _0.027_ | 0.059 | 0.191 | 0.033 | 0.029 | 0.028 | _0.027_ | _0.027_ | 0.029 | 0.028 | _0.027_ | **0.026** |
| Bandgap (OPT) | 0.122 | 0.239 | 0.265 | 0.137 | 0.127 | 0.124 | 0.122 | 0.120 | 0.120 | 0.122 | **0.116** | 0.118 |
| Total Energy | 0.029 | 0.074 | 0.486 | 0.035 | 0.032 | 0.032 | 0.029 | _0.028_ | 0.029 | 0.032 | _0.028_ | **0.027** |
| Ehull | 0.047 | 0.142 | 0.286 | 0.064 | 0.055 | 0.047 | 0.044 | _0.033_ | 0.045 | 0.040 | 0.036 | **0.032** |
| Bandgap (MBJ) | 0.264 | 0.394 | 0.649 | 0.300 | 0.270 | 0.282 | 0.261 | 0.251 | 0.280 | 0.256 | **0.240** | _0.242_ |
| Bulk Modulus (Kv) | 9.382 | 10.50 | 12.54 | 11.21 | 10.11 | 10.79 | 9.617 | 9.546 | 9.818 | 9.140 | _9.099_ | **8.817** |
| Shear Modulus (Gv) | 8.982 | 10.00 | 14.77 | 10.76 | 9.232 | 9.826 | 9.098 | _8.808_ | 9.108 | 9.422 | 8.966 | **8.528** |
| SLME (%) | 4.546 | 5.291 | 6.133 | 5.260 | 4.570 | 4.610 | 4.583 | _4.300_ | 4.636 | 4.415 | 4.529 | **4.256** |
| Spillage | 0.367 | 0.374 | 0.361 | 0.398 | 0.361 | 0.373 | 0.360 | 0.358 | _0.349_ | 0.362 | **0.340** | **0.340** |
| Formation Energy | 0.050 | 0.049 | 0.405 | 0.021 | 0.019 | 0.018 | 0.018 | _0.016_ | 0.019 | 0.017 | 0.018 | **0.015** |
| Bandgap (OPT) | 0.204 | 0.392 | 0.565 | 0.211 | 0.204 | 0.202 | 0.193 | 0.189 | _0.188_ | 0.195 | 0.191 | **0.184** |
| Bulk Modulus (Kv) | 0.046 | 0.041 | 0.055 | 0.043 | 0.040 | 0.042 | 0.038 | 0.034 | 0.038 | _0.036_ | 0.037 | **0.032** |
| Shear Modulus (Gv) | 0.070 | 0.068 | 0.075 | 0.073 | 0.065 | 0.073 | 0.064 | _0.062_ | 0.066 | 0.069 | 0.063 | **0.059** |

Table 2: Summary of MAE results for additional baseline models on various properties on JARVIS-DFT (top) and Materials Project (bottom).The best and second-best performances are shown in bold and underlined, respectively.

the downstream crystal property prediction task, we use two benchmark (property labelled) crystal datasets: Materials Project (MP-2018.6.1) (Chen et al., 2019) and JARVIS-DFT (Choudhary et al., 2020). The JARVIS dataset is a widely used benchmark dataset with 55,722 crystal structures with corresponding properties derived from DFT-based calculations. Following prior state-of-the-art studies, we focus on nine properties for downstream prediction: formation energy, bandgap (OPT), bandgap (MBJ), total energy, bulk modulus, shear modulus, energy above hull (ehull), spillage, and SLME. The Materials Project is another benchmark dataset containing 69,239 materials with crystal structures and their calculated properties. Following the previous SOTA algorithms, here we choose four crystal properties, namely, formation energy, bandgap (OPT), bulk modulus (Kv), and shear modulus (Gv), respectively. Among these, formation energy and bandgap are available for all 69,239 crystals, while bulk and shear moduli are labeled for only 5,451 structures.

## 5.2 DOWNSTREAM TASK EVALUATION

### 5.2.1 RESULTS ON DFT BENCHMARK DATASETS

**Setup.** First, we evaluate the performance of our proposed CrysLDNet on crystal property prediction tasks using DFT datasets. To this end, we consider the aforementioned benchmark datasets: Materials Project (MP) and JARVIS-DFT, selecting four and nine properties, respectively. We follow prior works for dataset splits. For JARVIS, we adopt an 80 / 10 / 10 split for training, validation, and testing across all properties, consistent with Matformer (Yan et al., 2022). For MP, we follow ALIGNN (Choudhary & DeCost, 2021) for formation energy and bandgap (OPT) with 60,000 / 5,000 / 4,239 crystals as train, validation, and test, and follow Matformer for bulk and shear moduli with 4,664 / 393 / 393 crystals in the respective splits.

**Baseline.** To evaluate the effectiveness of our proposed CrysLDNet, we compare its performance with a diverse set of state-of-the-art models on crystal property prediction tasks. Specifically, we consider six supervised (train-from-scratch) models: CGCNN (Xie & Grossman, 2018), SchNet (Schütt et al., 2018), MEGNet (Chen et al., 2019), GATGNN (Louis et al., 2020), ALIGNN (Choudhary & DeCost, 2021), and Matformer (Yan et al., 2022), all trained from scratch

directly on property-labeled data. In addition, we benchmark CrysLDNet against five pretrained models for crystal materials: CrysXPP (Das et al., 2022), Crystal Twins (Magar et al., 2022), CrysGNN (Das et al., 2023b), CrysDiff (Song et al., 2024), and DPF (Shen et al., 2025a), which adopt a pretraining stage followed by finetuning. We report the Mean Absolute Error (MAE) between predicted and ground truth values on the test set in Table 1. For fairness and to avoid performance degradation due to insufficient hyperparameter tuning, for supervised models we directly use the baseline results reported in their respective papers.

**Results.** We derive several insightful observations from the results in Table 1. First, compared to CGCNN, prior self-supervised pretrained models such as CrysXPP, CrysGNN, and Crystal Twins show clear improvements across most properties by effectively leveraging pretrained knowledge. However, advanced train-from-scratch models like ALIGNN and Matformer outperform them. Next, diffusion-based pretraining frameworks such as CrysDiff and DPF surpass these strong supervised models, further reducing the error. Finally, our proposed latent diffusion–based pretraining framework, CrysLDNet achieves additional improvements over both CrysDiff and DPF. Overall, CrysLDNet consistently outperforms all baseline models, both supervised and pretrained, across all properties in both datasets, with average improvements of 6.93% on JARVIS and 7.83% on MP over the second-best baseline. These results highlight the effectiveness of latent diffusion–based pretraining, which, by operating in a smoother latent space, captures the underlying chemical and structural semantics. This enables the learning of richer and more expressive crystal representations, thereby improving property prediction accuracy during subsequent finetuning.

**Results on New Baselines.** We have further expanded our experimental evaluation by incorporating several recent baseline models, including CoGN (Ruff et al., 2024), DimeNet++ (Gasteiger et al., 2020), Equiformer (Liao & Smidt, 2023), PotNet (Lin et al., 2023), eComformer (Yan et al., 2024), iComformer (Yan et al., 2024), and PDDformer (Shen et al., 2025b), all of which yield better result than Matformer. In addition, we developed improved variants of CrysLDNet using eComformer, iComformer, and PDDformer as backbones. The performance of these models across different JARVIS and MP properties is provided in Table-2. Importantly, these enhanced CrysLDNet variants outperform all baseline models across all evaluated properties on both the JARVIS and MP datasets. This result demonstrates that our pretraining framework is not only effective but also fully compatible with different backbone architectures. Consequently, any future, more powerful transformer models can be integrated into our pretrain–finetune pipeline without modification, and we expect them to yield even stronger performance.

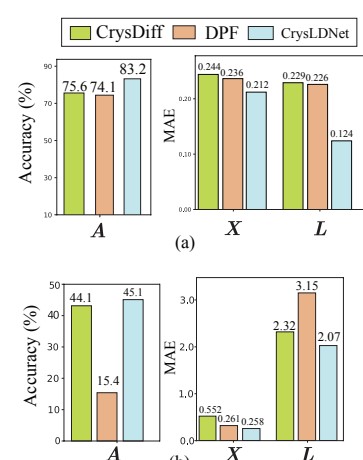

Figure 3: Reconstruction performance of pretrained embeddings of CrysDiff, DPF and CrysLDNet on (a) Genome dataset and (b) Jarvis dataset.

**Expressiveness of Latent Representations.** We conducted an additional experiment to assess the expressiveness of the representations learned through our latent diffusion–based pretraining, compared with approaches that apply diffusion directly in feature space. For each material, we passed it through the pretrained encoders of CrysDiff, DPF, and CrysLDNet (pretrained using Genome dataset) and evaluated their ability to reconstruct atom types, coordinates, and lattice parameters. This evaluation was performed on both the Genome and JARVIS datasets. Figure 3 reports atom-type accuracy as well as MAE for coordinate and lattice reconstruction. On the Genome dataset, all methods achieve high accuracy for $A$ and low MAE for $X$ and $L$, suggesting that reconstruction is relatively straightforward when the evaluation distribution aligns with the pretraining distribution. In contrast, performance on the JARVIS dataset—where none of the models were pretrained—reflects a zero-shot setting, and the distribution shift leads to a moderate decline in reconstruction quality, as expected. Across both datasets, the embeddings produced by CrysLDNet show consistently strong performance relative to the baselines in all reconstruction tasks. This indicates that latent diffu-

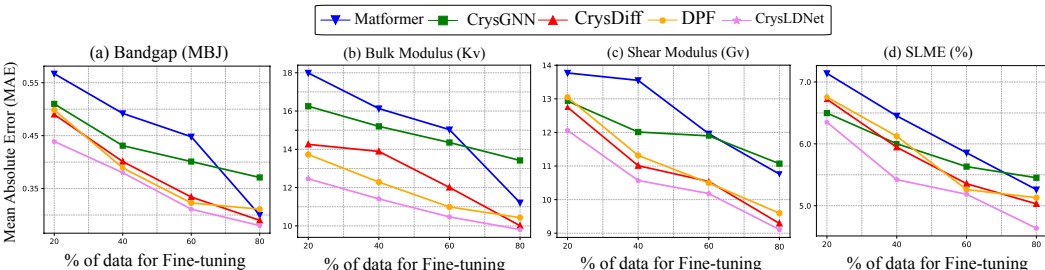

Figure 4: The prediction performance (MAE) of four properties on the JARVIS-DFT dataset with sparse data. In particular, for 20% fine-tuning data percentage improvement our model CrysLDNet with respect to the second best model are 10.41%, 9.26%, 5.39%, and 5.56%, respectively.

| Setup | Formation Energy | Bandgap (OPT) | Total Energy | Ehull | Bandgap (MBJ) | Bulk Modulus (Kv) | Shear Modulus (Gv) | SLME (%) | Spillage |
|---|---|---|---|---|---|---|---|---|---|
| VAE only | 0.031 | 0.126 | 0.032 | 0.059 | 0.284 | 10.61 | 9.773 | 4.970 | 0.374 |
| LDM only | 0.030 | 0.123 | 0.031 | 0.052 | 0.302 | 10.37 | 9.815 | 4.878 | 0.370 |
| Only A | 0.032 | 0.125 | 0.031 | 0.058 | 0.285 | 10.49 | 9.418 | 4.673 | 0.355 |
| Only X | 0.031 | 0.122 | 0.030 | 0.060 | 0.294 | 10.21 | 9.336 | 4.853 | 0.352 |
| Only L | 0.032 | 0.136 | 0.032 | 0.055 | 0.292 | 10.46 | 9.621 | 4.851 | 0.351 |
| Both A,X | 0.034 | 0.125 | 0.031 | 0.052 | 0.286 | 10.25 | 9.268 | 4.789 | 0.358 |
| Both A,L | 0.032 | 0.127 | 0.030 | 0.051 | 0.288 | 10.42 | 9.272 | 4.709 | 0.359 |
| Both L,X | 0.033 | 0.124 | 0.029 | 0.046 | 0.291 | 10.51 | 9.305 | 4.678 | 0.354 |
| CrysLDNet | **0.029** | **0.120** | **0.029** | **0.045** | **0.280** | **9.818** | **9.108** | **4.636** | **0.349** |

Table 3: Results of ablation studies on the JARVIS dataset, conducted to examine the impact of different pretraining components of CrysLDNet.

sion–based pretraining yields representations that capture meaningful structural information. Since crystal properties depend on atomic arrangement and lattice geometry, such expressive embeddings are well suited for supporting downstream prediction tasks.

### 5.2.2 RESULTS ON LIMITED TRAINING DATA

Next, we investigate the effectiveness of our proposed pretraining framework under limited-data settings. Among the properties reported in Table 1, the Bulk and Shear Modulus in the Materials Project dataset have relatively few training samples (only 5,451 structures). We observe that CrysLDNet outperforms all baseline models by a significant margin, achieving 9.52% and 9.59% improvements over the second-best model for Bulk and Shear Modulus, respectively. Furthermore, we conduct additional experiments by varying the amount of labeled training data available during finetuning. Specifically, we vary the proportion of training data to 20%, 40%, 60%, & 80%, and evaluate their performance on the remaining test set. For comparison, we select four strong baselines: Matformer, CrysGNN, CrysDiff, and DPF. The results are reported in Figure 4. Firstly, we observe that compared to Matformer, all pretrained models achieve better performance in limited data settings, such as when only 20%, 40%, or 60% of the labeled training data is available. This highlights the strength of the pretrain–finetune framework. Moreover, among the diffusion-based pretraining frameworks, our proposed CrysLDNet model consistently outperforms the other baselines across all training data ratios. This demonstrates the richness of the representations learned through latent diffusion pretraining, which makes the model highly robust in sparse data regimes.

### 5.3 ABLATION STUDY

The pretraining of CrysLDNet involves a Variational Autoencoder and a Latent Diffusion Model. To better understand the contribution of each component we analyze their effects on downstream property prediction tasks. Specifically, we design two ablation experiments: (a) *VAE Only* and (b) *LDM Only*. In VAE Only case, we employ only the VAE component using the loss in Eq. 3, and then finetune the resulting encoder on the downstream tasks. The LDM-Only model does not include a VAE, instead, it uses a MatFormer encoder with randomly initialized parameters to produce latent representations, and the latent diffusion model operates on that. Both the LDM and encoder are jointly trained from scratch. The key difference between the second stage of the full CrysLDNet pipeline and the LDM-Only baseline lies in the state of the encoder feeding the latent diffusion model. In the full CrysLDNet pipeline, the VAE is first pretrained using the reconstruction loss

| Epoch | Bulk Modulus (Kv) | Shear Modulus (Gv) | SLME (%) | Spillage |
|---|---|---|---|---|
| 5 | 10.35 | 9.674 | 4.740 | 0.356 |
| 10 | 10.27 | 9.655 | 4.695 | 0.355 |
| 20 | 10.16 | 9.492 | 4.651 | 0.352 |
| 30 | 9.912 | 9.369 | 4.647 | 0.351 |
| 50 | **9.818** | **9.108** | **4.636** | **0.349** |

Table 4: Ablation Studies on VAE latent on epochs CrysLDNet.

(Eq. 3), and the resulting encoder is then further refined during the latent diffusion training. Thus, the LDM operates on a pretrained and semantically meaningful latent space. Both the encoder and the denoising network are trained jointly from scratch, and the encoder is further finetuned for downstream tasks. We report the results for JARVIS-DFT dataset in Table 3 (Top). Across both setups, we observe performance degradation for all properties, highlighting the importance of incorporating both modules during pretraining. Notably, pretraining with only the LDM achieves comparatively lower errors than only the VAE. This shows the role of diffusion models in capturing richer representations, compared to unsupervised pretraining with only the VAE. Further, in the VAE pretraining phase, we jointly reconstruct $A$, $X$ and $L$. To understand the impact of each of these reconstruction objectives, we conduct an ablation study where only subsets of atom types, coordinates, and lattice structures are reconstructed. The results, reported in Table 3 (Bottom), show performance degradation compared to CrysLDNet, indicating that reconstructing all three $(A, X, L)$ during VAE pretraining leads to better performance.

### 5.4 ANALYSIS OF THE LATENT SPACE LEARNED BY THE ENCODER.

We measured the Mutual Information (MI) between the learned latent representations and the underlying material structure to quantify how much structural information the encoder retains (Hjelm et al., 2018), comparing the VAE-only model with CrysLDNet (VAE+LDM). CrysLDNet exhibits substantially higher MI with both atom types (VAE: 3.0906 → CrysLDNet: 4.5465) and atomic coordinates (VAE: 1.3124 → CrysLDNet: 2.4864), indicating that the LDM refinement produces richer and more structurally grounded embeddings.

We posit that more expressive latent representations directly translate into improved downstream property-prediction performance. This is consistent with the results in Table 3, where CrysLDNet outperforms the VAE-only baseline. To examine this effect in greater detail, we performed an ablation study in which VAE latents were progressively refined by the LDM for varying numbers of training epochs. We observed that, with increasing epochs, the LDM consistently enhanced the VAE representations, leading to monotonic improvements in property-prediction accuracy. We report the representative results on four properties from the JARVIS dataset, present in Table 4.

## 6 PRETRAINING COMPLEXITIES

Our two-stage pretraining (VAE + LDM) is slightly more expensive than a single-stage model like DPF. For clarity, we report the training times on an L40 GPU server. DPF requires $\approx 377$ minutes (6.28 GPU-hours) in total, while our VAE and LDM stages take $\approx 210$ minutes (3.5 GPU-hours) and 301 minutes (5.02 GPU-hours) respectively, for a total of 511 minutes (8.52 GPU-hours). We have reported more details in Appendix (Table-8). Overall, this cost remains manageable in practice and is only incurred once during pretraining.

## 7 CONCLUSION

In this work, we propose CrysLDNet, a novel latent diffusion–based pretraining framework for crystal property prediction. CrysLDNet learns robust and enriched crystal representations through VAE encoding and latent diffusion denoising, enabling efficient and accurate downstream property prediction. Extensive experiments on widely used DFT datasets demonstrate that CrysLDNet outperforms both training-from-scratch and pretrained baselines by a significant margin. Furthermore, the learned representations are robust in sparse data regimes and sufficiently expressive to mitigate DFT error bias when finetuned with limited experimental data. The pretraining framework can be extended beyond structural graph information in a multi-modal setting to incorporate other relevant data, such as text and images. More broadly, this approach opens avenues for applying diffusion models to other graph-related tasks.

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

LATENT DIFFUSION PRETRAINING FOR CRYSTAL PROPERTY PREDICTION (TECHNICAL APPENDIX)

THE USE OF LARGE LANGUAGE MODELS (LLMS)

No Large Language Models (LLMs) were used in conducting the research presented in this paper. However, we employed an LLM (ChatGPT) solely for editorial purposes, including refining grammar, spelling, word choice, and overall clarity of the manuscript.

## A  SYMMETRY IN CRYSTAL STRUCTURE

Crystal materials satisfy physical symmetry properties Dresselhaus et al. (2007); Zee (2016), one of the major challenges is the learned representation must satisfy invariance w.r.t. translation, rotation, and periodic transformations.

- **Rotational Invariance :** If we rotate the atom coordinates and lattice matrix, the material remains unchanged. Formally, using any orthogonal rotational matrix $\mathbf{Q} \in R^{3\times3}$ (satisfying $\mathbf{Q}^T\mathbf{Q} = \mathbf{I}$), if we rotate $X$ and $L$ of any material $M$ and generate new $M_R = (A, QX, QL)$, then actually different representations of the same material.
- **Periodic Translation Invariance :** If we translate the atom coordinates by a random vector it will not change the structure of the material. Also, since the atoms in the unit cell can periodically repeat itself infinite times along the lattice vector, there can be many choices of unit cells and coordinate matrices representing the same material. Formally, given any material $M = (A, X, L)$, if we translate $X$ by an arbitrary translation vector $\mathbf{u} \in \mathbb{R}^3$ and if there exist a function $w[X] := X - \lfloor X \rfloor$, new generated material $M_P = (A, w(X + \mathbf{u1}^T), L)$ will be the same as $M$. Hence the structure of a crystal remains the same when applying periodic translation to $X$.

## B  RELATED WORK

### B.1  DIFFUSION MODELS

The fundamental idea of the diffusion model, as initially proposed by (Sohl-Dickstein et al., 2015), is to gradually corrupt data with diffusion noise and learn a neural model to recover back data from noise. Idea of diffusion further developed in two broad categories - 1) *Score Matching Network* (Song & Ermon, 2019; 2020) and 2) *Denoising Diffusion Probabilistic Models (DDPM)* (Ho et al., 2020). In recent times diffusion models have emerged as a powerful new family of deep generative models, achieving remarkable performance records across numerous applications such as image synthesis (Dhariwal & Nichol, 2021), molecular conformer generation (Shi et al., 2021; Xu et al., 2022), molecular graph generation (Liu et al., 2021), protein folding (Wu et al., 2021; Luo et al., 2022) etc. Recently, several studies have successfully developed latent diffusion models (LDMs) with promising results across various applications, including image generation (Vahdat et al., 2021), point clouds (Vahdat et al., 2022), and text generation (Li et al., 2022). One of the most remarkable successes among them is the Stable Diffusion (Rombach et al., 2022) models, which demonstrate surprisingly realistic text-guided image generation results.

### B.2  CRYSTAL PROPERTY PREDICTION

In recent times, graph neural network(GNN) Xie & Grossman (2018); Chen et al. (2019); Louis et al. (2020); Park & Wolverton (2020); Schmidt et al. (2021); Choudhary & DeCost (2021); Yan et al. (2022); Lin et al. (2023) based approaches become popular tools for crystal property prediction. Earlier approaches Xie & Grossman (2018); Chen et al. (2019); Louis et al. (2020); Park & Wolverton (2020); Schmidt et al. (2021) construct a multi-edge graph from the 3D crystal structure and apply a GNN model to encode the neighborhood structural information around an atom. Building on this, numerous studies have proposed various GNN architecture variants that integrate domain-specific knowledge into the encoder to improve crystal representation learning. ALIGNN Choudhary & DeCost (2021) incorporates bond angular information among edges to capture many-body interactions; whereas Matformer Yan et al. (2022) is designed to be invariant to periodicity, enabling it to explicitly capture repeating patterns. Moreover, CrysMMNet Das et al. (2023a) leverages multi-modal information where they fuse textual description with crystal graph structure to enhance the property prediction.

Data scarcity remains a significant challenge in this field, motivating the development of various graph pretraining strategies. CrysXPP (Das et al., 2022) introduced unsupervised pretraining fol-

lowed by fine-tuning on property-labeled data, while CrysGNN (Das et al., 2023b) extended this idea through large-scale self-supervised pretraining, distilling knowledge from unlabeled crystal structures and transferring it to diverse property predictors to improve accuracy. Similarly, Crystal Twins (Magar et al., 2022) applied self-supervised learning to pretrain the CGCNN encoder using the Barlow Twins loss. More recently, diffusion-based pretraining frameworks have been explored for crystal structure reconstruction. CrysDiff (Song et al., 2024) employs a joint denoising diffusion model to reconstruct crystal structures from atomic compositions during pretraining, and during fine-tuning it conditions diffusion on target property values while keeping structures fixed. In contrast, DPF (Shen et al., 2025a) perturbs atom types, positions, and lattice constants during pretraining, reconstructs the native crystal structure, and finetunes the learned representation for downstream property prediction.

### B.3 Crystal Material Generation

In the past, there were limited efforts in creating novel periodic materials, with researchers concentrating on generating the atomic composition of periodic materials while largely neglecting the 3D structure. With the advancement of generative models, the majority of the research focuses on using popular generative models like VAEs or GANs to generate 3D periodic structures of materials, however, they either represent materials as three-dimensional voxel images (Court et al., 2020; Hoffmann et al., 2019; Long et al., 2021; Noh et al., 2019) and generate images to depict material structures (atom types, coordinates, and lattices), or they directly encode material structures as embedding vectors (Kim et al., 2020; Ren et al., 2020; Zhao et al., 2021). However, these models neither incorporate stability in the generated structure nor are invariant to any Euclidean and periodic transformations. Recent advancements in equivariant diffusion models have opened up a promising trajectory for the generation of novel three-dimensional periodic structures of crystal materials. CDVAE (Xie et al., 2021) was the first work that integrated a variational autoencoder (VAE) and powerful score-based decoder network, work directly with the atomic coordinates of the structures and uses an equivariant graph neural network to ensure euclidean and periodic invariance. Subsequently, numerous studies (Luo et al., 2023; Jiao et al., 2023; Zeni et al., 2023; Jiao et al., 2024; Yang et al., 2023; Das et al., 2025) have utilized diffusion models to learn the joint distribution of atom types, coordinates, and lattice structures, enabling the generation of stable periodic structures for novel materials. However, a key limitation of these diffusion-based models is that they operate directly in high-dimensional feature spaces, jointly modeling atom types, fractional coordinates, and lattice structures. The complexity of handling these heterogeneous components demands a highly sophisticated denoising architecture and typically requires many diffusion steps to produce high-quality crystal structures. Recently, CrysLDM (Khastagir et al.) and ADiT (Joshi et al., 2025) addressed this issue by introducing latent diffusion models that operate in a smoother, lower-dimensional latent space, enabling the generation of more stable and valid materials.

## C  Joint VAE–Flow Training Stability

Joint VAE–Flow training can become unstable and may cause the encoder to collapse, but this usually happens when the encoder is trained from scratch with random initialization. A common and effective solution is to pretrain the encoder so that it starts from a meaningful state. In our case, the encoder is not trained from scratch, rather we first pretrain the encoder using a VAE with a reconstruction loss over atom types, coordinates, and lattice, along with a KL regularizer (Eq. 3). Only after this step, the encoder is further refined during joint training with the LDM (Algorithm 1, Stage 2). This warm start with VAE ensures that the encoder already produces meaningful, non-collapsed latent representations, which the LDM then improves rather than driving toward a trivial constant output.

However, to investigate this phenomenon further, we compared the losses of (i) a standard LDM trained without our pretrained encoder(without VAE) and (ii) our full CrysLDNet model, and the results for 50 epochs are reported in Fig-5.

In the first setting, the loss collapses almost immediately: it drops from $1.43 \rightarrow 0.16 \rightarrow 0.07 \rightarrow 0.03$ within the first four epochs, and reaches the order of $10^{-3} - 10^{-4}$ by epoch 12 e.g., 0.00148 at epoch 12 and (0.00063) at epoch 20). By epoch 50, the loss falls to $1.7 \times 10^{-4}$, indicating convergence to a near-trivial solution. Such extremely rapid loss decay is consistent with the encoder collapsing to an almost constant latent representation, allowing the flow network to minimize the matching loss trivially.

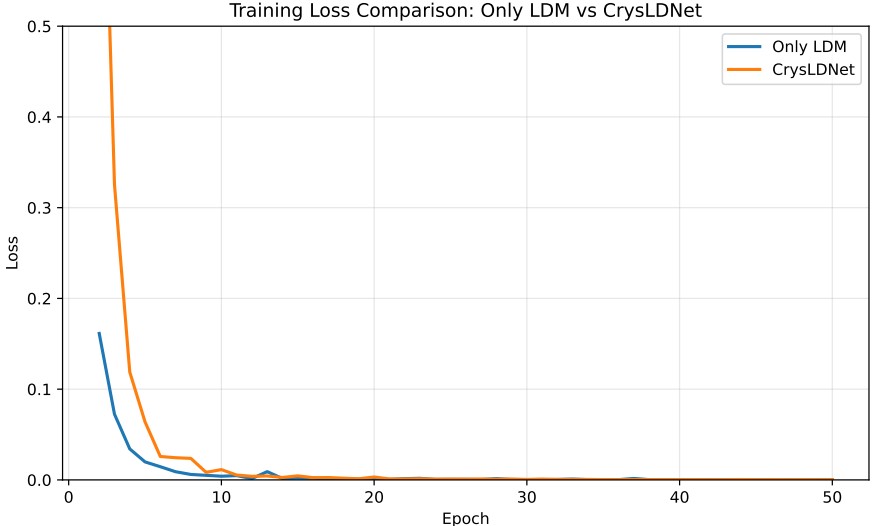

Figure 5: Comparison of loss curves between LDM (Without VAE Loss) and CrysLDNet.

In sharp contrast, CRYSLDNET does not exhibit this behavior. Its loss decreases much more gradually—from (5.39) (epoch 1) to (0.88), (0.32), and (0.11) in the first four epochs—and stabilizes in the range of $10^{-3} - 10^{-2}$ during mid-training (e.g., (0.0083) at epoch 9, (0.0039) at epoch 12, (0.0026) at epoch 17). Even at epoch 20, the loss remains at (0.00319), which is significantly higher than the collapsed LDM baseline ((0.00063)), reflecting a non-collapsed and more expressive encoder.

This comparison clearly shows that collapse occurs only when the encoder lacks reconstruction or KL constraints (LDM without VAE), whereas the full CrysLDNet training remains stable and avoids the degenerate constant-latent solution the reviewer described.

# D    EXPERIMENTAL SETUP

## D.1    IMPLEMENTATION/ HYPERPARAMETERS DETAILS

All experiments are conducted on shared servers equipped with NVIDIA L40 GPUs. We pre-train the first stage for 50 epochs and the second stage for another 50 epochs using the AdamW optimizer with a batch size of 256, a learning rate of 1e-3, weight decay of $10^{-5}$, and a one-cycle scheduler. We then fine-tune the model on downstream crystal property prediction tasks for 1000 epochs with a batch size of 32, while keeping all other hyperparameters identical to those used during pre-training.

### D.1.1    BASELINE MODELS

To evaluate the effectiveness of our proposed CrysLDNet, we compare its performance with a diverse set of state-of-the-art models on crystal property prediction tasks. Specifically, we consider six supervised (train-from-scratch) models—CGCNN (Xie & Grossman, 2018), SchNet (Schütt et al., 2018), MEGNet (Chen et al., 2019), GATGNN (Louis et al., 2020), ALIGNN (Choudhary & DeCost, 2021), and Matformer (Yan et al., 2022), all trained directly on property-labeled data. In addition, we benchmark CrysLDNet against five pretrained models for crystal materials—CrysXPP (Das et al., 2022), Crystal Twins, CrysGNN (Das et al., 2023b), CrysDiff (Song et al., 2024), and DPF (Shen et al., 2025a), which adopt a pretraining stage followed by finetuning.

### D.1.2    EVALUATION METRICS

Following prior works (Choudhary & DeCost, 2021; Xie & Grossman, 2018), we use the Mean Absolute Error (MAE) between predicted and true property values to evaluate the performance of all baseline models on the crystal property prediction task.

$$\text{MAE}(\mathcal{M}, f) = \frac{1}{m} \sum_{i=1}^{m} |f(\mathcal{M}_i) - y_i|$$

| Metric | $|C|$ | $|A|$ | $|T|$ | Volume | Density |
|--------|------|------|------|--------|---------|
| Max    |        | 40   | 6 | 9291.69 | 24.10 |
| Min    | 380740 | 2    | 2 | 25.84   | 0.18  |
| Mean   |        | 4.10 | 4 | 436.96  | 8.34  |
| Var    |        | 0.46 | 0 | 62283.63 | 7.38 |

Table 5: Statistics of the GNoME dataset. Max, Min, Mean, and Var denote the maximum, minimum, average, and variance, respectively. $|A|$, $|T|$, and $|C|$ represent the number of atoms per crystal, the number of atom types per crystal, and the total number of crystal structures.

### D.1.3 RESULTS USING EXPERIMENTAL DATA

A key challenge in materials science is the scarcity of experimental data for crystal properties Kubaschewski et al. (1993); Bracht et al. (1995); Turns (1995), which prevents models from achieving experimental-level predictive accuracy. To compensate, most existing approaches rely on DFT-tagged data for training (As shown in Section 5.2.1). However, the inherent approximations in

| Experiment Setup | CryXPP | CrysGNN | CrysDiff | DPF | CrysLDNet |
|------------------|--------|---------|----------|-----|-----------|
| Finetune on DFT Test on Expt | 0.307 | 0.253 | 0.211 | 0.217 | **0.205** (2.84) |
| Finetune on DFT + 20% Expt Test on 80% Expt | 0.158 | 0.135 | 0.102 | 0.109 | **0.097** (4.90) |
| Finetune on DFT + 80% Expt Test on 20% Expt | 0.110 | 0.096 | 0.087 | 0.070 | **0.068** (2.86) |

Table 6: Comparison of experimental settings across baselines and our method. Reported values are MAE.

DFT calculations introduce systematic errors when compared with experimental measurements, leading to DFT error bias. Current train-from-scratch methods can hardly mitigate such DFT error bias. However, CrysXPP have demonstrated that pretraining followed by fine-tuning on limited experimental data can help mitigate this issue, and subsequent works like CrysGNN and CrysDiff have reinforced this observation. Motivated by these insights, we examine whether our proposed framework can further alleviate DFT error when fine-tuned on the small pool of available experimental data. Specifically, we consider the OQMD-EXP dataset Kirklin et al. (2015), which contains approximately 1,500 crystal materials with experimentally measured formation energies. All models are first trained on formation energy using DFT data, and we evaluate them in three setups. In the first, they are tested directly on the complete experimental dataset in a zero-shot manner. In the second, they are finetuned on 20% of the experimental dataset and tested on the remaining 80%. In the third, they are finetuned on 80% of the experimental dataset and tested on the remaining 20%. We report the MAE for all these settings of different competing baseline models in Table 6. We observe that as the amount of experimental training data increases, all the SOTA models consistently achieve lower errors. Moreover, CrysLDNet outperforms them across all three setups, demonstrating its strong expressiveness in mitigating DFT error bias.

### D.1.4 STATISTICAL SIGNIFICANCE OF THE RESULTS

We perform a comprehensive statistical analysis to ensure the robustness and reliability of the reported performance improvements. For each variant of CrysLDNet, we conduct five independent runs with different random seeds and report the mean, standard deviation, and 95% confidence interval (CI). In addition, for each backbone, we compute paired t-test p-values to assess the statistical significance of the improvements. Specifically, we select PDDFormer, iComFormer, eComFormer, and Matformer, along with the corresponding variants of CrysLDNet that use these encoders as backbones, to evaluate statistical significance. The complete results on JARVIS dataset are presented in Table 7.

These statistical measures demonstrate that the performance gains achieved by CrysLDNet are consistent and reproducible across multiple runs. Notably, the paired t-tests yield p-values below 0.05 for most evaluated properties, confirming that the improvements are statistically significant. Overall, this analysis verifies that the superiority of CrysLDNet does not arise from random variation but reflects genuine and meaningful performance improvements across downstream tasks.

| Property | PDDFormer | CrysLDNet (PDDFormer) | | | iComFormer | CrysLDNet (iComFormer) | | |
|---|---|---|---|---|---|---|---|---|
| | | Mean±Std | CI | P-Value | | Mean±Std | CI | P-Value |
| Formation Energy | 0.027 | 0.026±0.0004 | (0.0255, 0.0265) | 0.005 | 0.0272 | 0.0270±0.0001 | (0.0269, 0.0271) | 0.011 |
| Bandgap | 0.120 | 0.118±0.001 | (0.117, 0.119) | 0.011 | 0.122 | 0.116±0.002 | (0.114, 0.118) | 0.003 |
| Total Energy | 0.028 | 0.027±0.0002 | (0.0268, 0.0272) | 0.0003 | 0.029 | 0.028±0.0003 | (0.0276, 0.0284) | 0.002 |
| Ehull | 0.033 | 0.032±0.0005 | (0.031, 0.033) | 0.011 | 0.044 | 0.036±0.003 | (0.032, 0.040) | 0.004 |
| mbj Bandgap | 0.251 | 0.242±0.004 | (0.237, 0.247) | 0.007 | 0.261 | 0.240±0.007 | (0.231, 0.249) | 0.003 |
| Bulk Modulus | 9.546 | 8.817±0.240 | (8.519, 9.115) | 0.0024 | 9.617 | 9.099±0.200 | (8.851, 9.347) | 0.004 |
| Shear Modulus | 8.808 | 8.528±0.100 | (8.404, 8.652) | 0.003 | 9.098 | 8.966±0.080 | (8.867, 9.065) | 0.020 |
| SLME | 4.300 | 4.256±0.010 | (4.244, 4.268) | 0.0006 | 4.583 | 4.529±0.020 | (4.504, 4.554) | 0.004 |
| Spillage | 0.358 | 0.340±0.007 | (0.331, 0.349) | 0.004 | 0.360 | 0.340±0.009 | (0.329, 0.351) | 0.007 |

(a) Comparison for PDDFormer and iComFormer backbones.

| Property | eComFormer | CrysLDNet (eComFormer) | | | Matformer | CrysLDNet (Matformer) | | |
|---|---|---|---|---|---|---|---|---|
| | | Mean±Std | CI | P-Value | | Mean±Std | CI | P-Value |
| Formation Energy | 0.0284 | 0.0280±0.0003 | (0.0276, 0.0284) | 0.040 | 0.0325 | 0.029±0.001 | (0.028, 0.030) | 0.001 |
| Bandgap | 0.124 | 0.122±0.001 | (0.121, 0.123) | 0.011 | 0.137 | 0.120±0.010 | (0.108, 0.132) | 0.019 |
| Total Energy | 0.0324 | 0.0320±0.0003 | (0.0316, 0.0324) | 0.041 | 0.035 | 0.029±0.002 | (0.027, 0.031) | 0.003 |
| Ehull | 0.047 | 0.040±0.004 | (0.035, 0.045) | 0.017 | 0.064 | 0.045±0.009 | (0.034, 0.056) | 0.010 |
| mbj Bandgap | 0.280 | 0.256±0.009 | (0.245, 0.267) | 0.004 | 0.300 | 0.280±0.010 | (0.268, 0.292) | 0.011 |
| Bulk Modulus | 10.79 | 9.140±1.200 | (7.650, 10.630) | 0.040 | 11.21 | 9.818±1.000 | (8.576, 11.060) | 0.035 |
| Shear Modulus | 9.826 | 9.422±0.200 | (9.174, 9.670) | 0.011 | 10.76 | 9.108±1.100 | (7.742, 10.474) | 0.028 |
| SLME | 4.610 | 4.415±0.090 | (4.303, 4.527) | 0.008 | 5.260 | 4.636±0.400 | (4.139, 5.133) | 0.025 |
| Spillage | 0.373 | 0.362±0.004 | (0.357, 0.367) | 0.003 | 0.398 | 0.349±0.010 | (0.337, 0.361) | 0.0004 |

(b) Comparison for eComFormer and Matformer backbones.

Table 7: Statistical comparison of *CrysLDNet* across four backbone models.

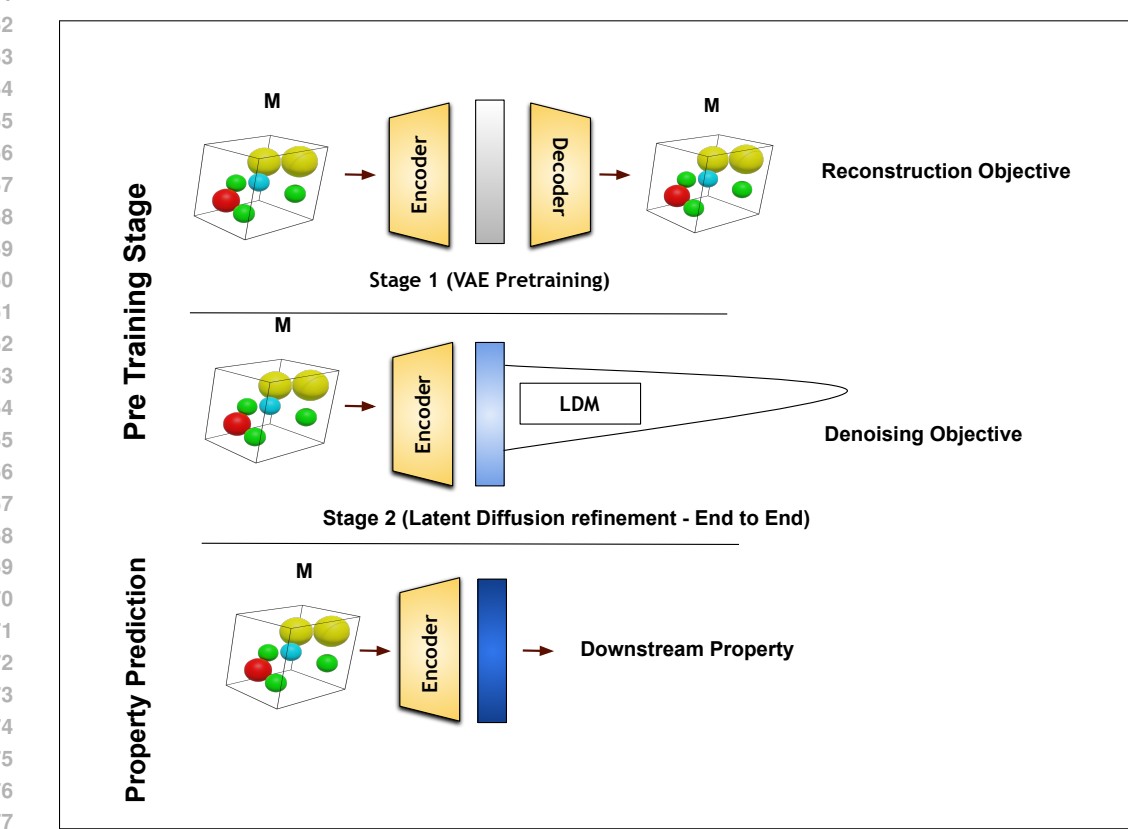

Figure 6: Simplified View of CrysLDNet Framework

| Parameters | DPF | VAE | LDM | CrysLDNet (Total) |
|---|---|---|---|---|
| Resources Used for Pre-Training | 1× NVIDIA L40 GPU server | 1× NVIDIA L40 GPU server | 1× NVIDIA L40 GPU server | 1× NVIDIA L40 GPU server |
| Memory | 15842 MB | 5986 MB | 8482 MB | 14468 MB |
| Total Training Time | $\approx 377$ min | $\approx 210$ min | $\approx 301$ min | $\approx 511$ min |
| GPU Hours (for Training) | $\approx 6.28$ h | $\approx 3.5$ h | $\approx 5.02$ h | $\approx 8.52$ h |

Table 8: Comparison of Computational Resources and Pre-Training Costs Across Models.

