# OpenReview forum: "Latent Diffusion Pretraining for Crystal Property Prediction"
_ICLR.cc/2026/Conference — Submitted to ICLR 2026_

### Official Review · Reviewer_Akui · 2025-10-28

**Soundness:** 2
**Presentation:** 3
**Contribution:** 1
**Rating:** 2
**Confidence:** 5

**Summary:**

This paper presents CrysLDNet, a generative pretraining approach that applies latent diffusion and denoising techniques to improve crystal property prediction. The authors aim to mitigate data sparsity and bias in chemical datasets by introducing a semi-supervised generative framework. The concept is well-motivated, as limited and unevenly distributed data are longstanding challenges in the field. The model integrates generative pretraining with downstream regression tasks and reports improvements across multiple benchmarks, suggesting enhanced generalization and robustness.

However, the experimental validation is limited by the choice of benchmarks and comparison baselines. The reported baselines are outdated and do not reflect the current state-of-the-art in crystal or molecular property prediction. For example, in formation energy prediction, recent models such as coGN (MAE = 0.0170) and DimeNet++ (MAE = 0.0235) outperform CrysLDNet by a noticeable margin. This raises concerns about the model’s competitive relevance. Moreover, the core architecture of CrysLDNet is relatively simple and appears to be a straightforward combination of existing denoising and autoencoding components, leaving little room for methodological novelty.

In addition, some claims in the manuscript require clarification. The authors refer to the model as a “Variational Autoencoder,” but the provided architecture and loss formulation resemble a standard autoencoder without explicit stochastic latent variables or KL regularization. Similarly, Figure 3 shows relatively low reconstruction accuracy (≈40%), which questions whether the latent representation sufficiently captures meaningful information. A correlation analysis between reconstruction fidelity and downstream prediction performance could strengthen the argument for the model’s effectiveness.

Overall, while CrysLDNet is simple and easy to follow, its contribution appears incremental, and its performance does not reach the level of more recent diffusion-based or graph-equivariant methods.

**Strengths:**

- Straightforward, interpretable model design.

- Demonstrates moderate improvement over the paper’s own baseline models.

- Addresses a relevant challenge in materials informatics — data sparsity and generalization.

**Weaknesses:**

- Reported benchmarks are outdated and exclude stronger recent baselines.

- Performance is not competitive with current state-of-the-art models.

- Architectural novelty is limited — appears as a simple combination of diffusion and autoencoder elements.

- Lack of clarity regarding the VAE structure and loss formulation.

- Insufficient analysis of reconstruction quality and computational efficiency.

**Questions:**

1. The model is described as a VAE, but there is no clear evidence of stochastic encoding or KL divergence losses. Could the authors clarify whether the model includes such components?

2. What specific denoising or diffusion steps are used in the latent pretraining? Are they based on DK losses or noise sampling as in typical diffusion models?

3. In Figure 3, reconstruction accuracy remains around 40%. How do the authors justify this as sufficient to capture key physical or structural features?

4. Could the authors provide a more detailed comparison to modern models such as coGN, ALIGNN, or Equiformer for a fair evaluation?

---

> ### Author Response · Authors · 2025-11-22
> **Point-by-point responses to Reviewer Akui(1/2)**
>
> We thank the reviewer for the valuable feedback. Below, we provide detailed point-by-point responses to each of the questions:
>
> - ***Clarification regarding VAE.***
> > We beg to differ with the reviewer’s observation that our architecture and loss formulation resemble a standard autoencoder without stochastic latent variables or KL regularization. As clearly detailed in Section 4.2.1 Variational Autoencoder (VAE), our model does implement the standard VAE framework. The loss function in Equation (3) explicitly contains both the reconstruction terms and the regularization term $L_{reg}$, which represents the KL divergence. These components together capture the stochastic encoding behavior expected in a VAE.
>
> - ***Clarification on Denoising and Diffusion Steps in Latent Pretraining.***
> >The complete formulation of this latent diffusion process, including noise injection, interpolation, vector-field construction, and the LDM loss, is fully detailed in Section 4.2.2 (Latent Diffusion Model) of the manuscript.
> >
> >In brief, we follow the standard diffusion-model formulation, but operate entirely in the latent space rather than in the feature space. Assuming the reviewer refers to Denoising KL Divergence (DK) losses, we clarify that our framework does not use DK loss. Instead, we adopt the conventional noise-sampling strategy used in typical diffusion models.
> >
> >We use a flow-matching framework in which noise is progressively added to the VAE latent embeddings, and the model learns to denoise them back to the clean latent space. During training, we linearly interpolate between a noisy latent sampled from a standard Gaussian distribution and the clean latent produced by the VAE encoder, generating intermediate noisy samples at randomly sampled time steps. A denoising network is then trained to approximate the true conditional vector field that maps these noisy latents back to the clean representations, optimized using an MSE loss between the predicted and ground-truth vector fields. This procedure effectively learns a continuous transformation from Gaussian noise to the true latent distribution.
>
> - ***Concerns for Reconstruction Accuracy and capturing key physical or structural features.***
> > We thank the reviewer for raising this concern. The lower atom-type reconstruction accuracy in Fig. 3 stems from the fact that both CrysLDNet and the baseline models are first pretrained on the Genome dataset, and the pretrained encoder is then evaluated directly on the JARVIS-DFT dataset to reconstruct atom types, coordinates, and lattice parameters. This evaluation is therefore a zero-shot setting, where the encoder has never encountered JARVIS-DFT structures during pretraining. Because the two datasets differ significantly in composition and structural distributions, a moderate reconstruction performance under such distribution shift is expected. We intentionally followed this cross-dataset evaluation to measure how well the learned representation generalizes to unseen materials.
> >
> >To further address the reviewer’s concern and verify that the encoder indeed captures meaningful structural information on seen data, we additionally evaluate reconstruction performance on the Genome dataset itself, i.e., under a matched training–evaluation distribution. In this setting, we observe the following reconstruction results:
> >
> >| Model | A(Accuracy) | X(MAE) | L(MAE)
> >|-|-|-|-|
> >| CrysDiff| 75.6%| 0.244 |0.229|
> >|DPF|74.1%|0.236|0.226|
> >|CrysLDNet|**83.2%**|**0.212**|**0.124**|
> >
> > There are two key insights:
> > - Substantial improvements in atom-type reconstruction accuracy, atomic coordinate reconstruction, and lattice structure prediction, demonstrating that the encoder learns to represent detailed structural features when evaluated in-domain.
> > - CrysLDNet’s latent embeddings outperform those of DPF and CrysDiff, confirming that latent diffusion–based pretraining yields richer and more expressive representations of both composition and structure.
> >
> >We have added these in the revised manuscripts

---

> > ### Author Response · Authors · 2025-11-22
> > **Point-by-point responses to Reviewer Akui(2/2)**
> >
> > - ***Regarding New Baselines.***
> > > The primary goal of this paper is to introduce a pretraining framework that is fully agnostic to the choice of backbone architecture. Any encoder—Transformer, EGNN, or GNN—can be integrated into our framework, and the downstream performance naturally reflects the expressive power of the chosen backbone. Since our closest competitor among diffusion-based pretrain–finetune approaches is DPF, which uses Matformer, we limited our initial experiments to architectures up to Matformer for a fair comparison.
> > >
> > > However, as requested by the reviewers, we have expanded our experimental evaluation in the revised manuscript by incorporating several newer and stronger baseline models, including CoGN, DimeNet++, Equiformer, PotNet, eComformer, iComformer, and PDDformer, all of which are more advanced than Matformer. In addition, we developed improved variants of CrysLDNet using eComformer, iComformer, and PDDformer as backbones.
> > The performance of these models across different JARVIS and MP properties is summarized as follows:
> > >
> > > **Table 1: JARVIS-DFT Results (MAE)**
> > >
> > >| Property | CoGN | DimeNet++ | Equiformer | Matformer | PotNet | eComformer | iComformer | PDDformer | CrysLDNet (Matformer) | CrysLDNet (eComformer) | CrysLDNet (iComformer) | CrysLDNet (PDDformer) |
> > >|-|------|-----------|------------|-----------|--------|------------|------------|-----------|------------------|--------------------|--------------------|-------------------|
> > >| Formation Energy | 0.027 | 0.059 | 0.191 | 0.033 | 0.029 | 0.028 | 0.027 | 0.027 | 0.029 | 0.028 | 0.027 | **0.026** |
> > >| Bandgap (OPT) | 0.122 | 0.239 | 0.265 | 0.137 | 0.127 | 0.124 | 0.122 | 0.120 | 0.120 | 0.122 | **0.116** | _0.118_ |
> > >| Total Energy | 0.029 | 0.074 | 0.486 | 0.035 | 0.032 | 0.032 | 0.029 | 0.028 | 0.029 | 0.032 | 0.028 | **0.027** |
> > >| Ehull | 0.047 | 0.142 | 0.286 | 0.064 | 0.055 | 0.047 | 0.044 | 0.033 | 0.045 | 0.040 | 0.036 | **0.032** |
> > >| Bandgap (MBJ) | 0.264 | 0.394 | 0.649 | 0.300 | 0.270 | 0.282 | 0.261 | 0.251 | 0.280 | 0.256 | **0.240** | _0.242_ |
> > >| Bulk Modulus (Kv) | 9.382 | 10.50 | 12.54 | 11.21 | 10.11 | 10.79 | 9.617 | 9.546 | 9.818 | 9.140 | 9.099 | **8.817** |
> > >| Shear Modulus (Gv) | 8.982 | 10.00 | 14.77 | 10.76 | 9.232 | 9.826 | 9.098 | 8.808 | 9.108 | 9.422 | 8.966 | **8.528** |
> > >| SLME (%) | 4.546 | 5.291 | 6.133 | 5.260 | 4.570 | 4.610 | 4.583 | 4.300 | 4.636 | 4.415 | 4.529 | **4.256** |
> > >| Spillage | 0.367 | 0.374 | 0.361 | 0.398 | 0.361 | 0.373 | 0.360 | 0.358 | 0.349 | 0.362 | **0.340** | **0.340** |
> > >
> > >**Table 2: Materials Project Results (MAE)**
> > >
> > >| Property | CoGN | DimeNet++ | Equiformer | Matformer | PotNet | eComformer | iComformer | PDDformer | CrysLDNet (Matformer) | CrysLDNet (eComformer) | CrysLDNet (iComformer) | CrysLDNet (PDDformer) |
> > >|---------|------|-----------|------------|-----------|--------|------------|------------|-----------|------------------|--------------------|--------------------|-------------------|
> > >| Formation Energy | 0.050 | 0.049 | 0.405 | 0.021 | 0.019 | 0.018 | 0.018 | 0.016 | 0.019 | 0.017 | 0.018 | **0.015** |
> > >| Bandgap (OPT) | 0.204 | 0.392 | 0.565 | 0.211 | 0.204 | 0.202 | 0.193 | 0.189 | 0.188 | 0.195 | 0.191 | **0.184** |
> > >| Bulk Modulus (Kv) | 0.046 | 0.041 | 0.055 | 0.043 | 0.040 | 0.042 | 0.038 | 0.034 | 0.038 | 0.036 | 0.037 | **0.032** |
> > >| Shear Modulus (Gv) | 0.070 | 0.068 | 0.075 | 0.073 | 0.065 | 0.073 | 0.064 | 0.062 | 0.066 | 0.069 | 0.063 | **0.059** |
> > >
> > >Importantly, these enhanced CrysLDNet variants outperform all baseline models across all evaluated properties on both the JARVIS and MP datasets. Looking ahead, any future, more powerful transformer models can be seamlessly integrated into our pretrain–finetune paradigm, and we expect them to further improve performance.
> > >
> > >The revised manuscript now includes these updated results.

---

> ### Comment · Reviewer_Akui · 2025-11-24
>
> Thank you to the authors for the swift and detailed response.
>
> The clarifications regarding the first three questions were helpful. In particular, I appreciate the explanation in Section 4.2.1, and I apologize for previously overlooking the KL term in the VAE formulation.
>
> That said, several important points remain unresolved:
>
> **(1) Backbone-agnostic claim remains unclear**
>
> The rebuttal asserts that the proposed method is "architecture-agnostic" and can incorporate any backbone. However this claim is not clearly stated nor explained in the main paper, and it remains unclear which specific components can be replaced by different backbones. (It is not demonstrated how the architecture pipeline integrates these backbones.. Maybe to the finetuning part..)
>
> Without a more explicit architectural diagram or description in the main text, it is still difficult to understand the mechanism by which arbitrary encoders are plugged into the framework.
>
> If backbone-agnosticism is a key feature, it needs to be clearly stated and explicitly supported (ideally with an illustration or algorithmic description).
>
> **(2) Lack of statistical analysis for newly added results**
>
> The additional baseline comparisons and updated results are appreciated.
> However, the performance improvements appear incremental, and the new tables do not report any statistical metrics such as standard deviations, confidence intervals, or variation across multiple runs.
>
> Given the relatively small margins between models, it is difficult to determine whether the improvements are statistically meaningful or fall within random variability. A proper statistical treatment would significantly strengthen the experimental claims.
>
> **(3) Question about baseline selection remains unresolved**
>
> If the model is indeed backbone-agnostic, it is still unclear why stronger and more recent models were not included in the original submission, especially when some of them (e.g., CoGN, DimeNet++, Equiformer) are widely recognized baselines in crystal modeling.
>
> The explanation that the architecture is agnostic does not fully address this absence. The inconsistency between the initial baseline choice and the updated extended baselines raises concerns regarding the fairness and completeness of the evaluation in the original submission.
>
> **(4) Computational cost concerns**
>
> Since the method relies on generative pretraining (VAE + latent diffusion), computational cost becomes a major factor—especially when the pretraining dataset is large. This is important because the reported improvements over baselines are relatively modest, yet generative pretraining can be expensive in time, memory, and energy, and the paper does not report wall-clock training cost, GPU hours, or resource requirements.
>
> Without a clear discussion of computational overhead, it is difficult to judge whether the approach offers a favorable trade-off between performance gain and resource expenditure.
>
> A comparison of computational efficiency—both pretraining and finetuning—would therefore be very helpful.
>
> The authors’ response addressed several technical questions, and I appreciate the effort in providing new experiments. Nevertheless, the four concerns above remain key issues affecting the clarity, completeness, and interpretability of the work.
>
> I hope these points can be clarified or addressed in subsequent revisions.

---

> > ### Author Response · Authors · 2025-11-26
> > **Responses to additional questions raised by Reviewer Akui(1/2)**
> >
> > We thank the reviewer for the valuable feedback. Below, we provide detailed point-by-point responses to each of the questions:
> >
> > - ***Regarding Backbone-Agnostic Design:***
> > >We thank the reviewer for pointing out the need for a clearer explanation of the architecture-agnostic design. In the revised manuscript we have added a dedicated paragraph titled **“Backbone-Agnostic Design” (Lines 287–295)**, where we have explicitly stated that CrysLDNet is fully agnostic to the choice of encoder backbone and clarified which components are interchangeable. We now clearly specify that any crystal-GNN, EGNN, or transformer-based encoder can be plugged into the framework without modifying the latent diffusion model, decoder, or downstream prediction heads. We have also modified an architectural diagram (Figure 2) in the main paper that highlights the encoder as an interchangeable module and written in the caption about it. This makes the modularity of the design explicit and easy to understand.
> > >
> > >We believe these additions directly address the reviewer’s concern and improve the clarity of the paper.
> >
> > - ***Regarding baseline Selection:***
> > >Our choice of baselines was primarily dataset-dependent. We followed widely adopted prior works such as ALIGNN and Matformer, which use the MP-2018 and JARVIS datasets for property prediction tasks. Within these datasets, Matformer represents the strongest and most relevant model, and our closest diffusion-based pretrain–finetune competitor, DPF, also employs Matformer. Hence, we restricted our experiments to architectures up to Matformer to ensure a fair and directly comparable evaluation. (Similar evaluation setup followed in DPF [1], Crysdiff [2], Matformer [3], CrysGNN [4], ALIGNN [5]). Note, models such as CoGN, DimeNet++, and Equiformer did not report results on these datasets in their original publications. Our goal was not to explore numerous encoder backbones, but rather to highlight the core contribution—that the benefits of latent diffusion pretraining remain consistent across architectures. Therefore, we maintained a balanced and focused baseline set to keep the emphasis on our central findings.
> > >
> > >However, we sincerely thank the reviewer for the constructive suggestion. In response, rather than reiterating our original justification, we promptly ran all the requested models, along with several additional ones, to broaden and strengthen the baseline set. We also developed new variants of CrysLDNet using stronger encoder backbones to ensure an even more rigorous evaluation.
> > >
> > >As anticipated, the core conclusions of the paper remain unchanged. Nevertheless, the expanded baselines offer a more comprehensive comparison and, in our view, make the paper richer.
> > >
> > >We again thank the reviewer for the helpful recommendation and appreciate ICLR’s policy that allows authors to incorporate additional results during the rebuttal phase.
> > >
> > >[1] A Denoising Pre-training Framework for Accelerating Novel Material Discovery. AAAI 2025.
> > >
> > >[2] A diffusion-based pre-training framework for crystal property prediction. AAAI 2024.
> > >
> > >[3] Periodic graph transformers for crystal material property prediction. NeurIPS 2022.
> > >
> > >[4] Crysgnn: Distilling pre-trained knowledge to enhance property prediction for crystalline materials. AAAI 2023
> > >
> > >[5] Atomistic line graph neural network for improved materials property predictions. npj Computational Materials. 2021.
> >
> > - ***Regarding Computational Cost.***
> > >
> > > We thank the reviewer for raising this concern. For clarity, we report the computational details of the pretraining stage for DPF and CrysLDNet as follows:
> > >
> > >| **Parameters** | **DPF** | **VAE** | **LDM** | **CrysLDNet (Total)** |
> > >|-|-|-|-|-|
> > >| **Resources Used for Pre-Training** | 1× NVIDIA L40 GPU server | 1× NVIDIA L40 GPU server | 1× NVIDIA L40 GPU server | 1× NVIDIA L40 GPU server |
> > >| **Memory** | 15842 MB | 5986 MB | 8482 MB | 14468 MB |
> > >| **Total Training Time** | ≈ 377 min | ≈ 210 min | ≈ 301 min | ≈ 511 min |
> > >| **GPU Hours (for Training)** | ≈ 6.28 h | ≈ 3.5 h | ≈ 5.02 h | ≈ 8.52 h |
> > >
> > >Although CrysLDNet incurs a moderately higher pre-training time, this cost is incurred only once and remains fully manageable in practice. We have added this comparison to the revised manuscript. We believe that the one-time additional computational overhead is minimal, especially relative to the scale of modern hardware typically used for large pre-trained models.  Moreover, downstream finetuning is lightweight and similar to DPF (Same encoder+several layers of MLP).
> > >
> > >We have updated these details in **Section 6.PRETRAINING COMPLEXITIES(Line - 522/23)** in the revised manuscript.

---

> ### Author Response · Authors · 2025-11-26
> **Responses to additional questions raised by Reviewer Akui(2/2)**
>
> ***Statistical analysis for results***
>
> To address the concern regarding statistical robustness, we perform a comprehensive statistical analysis to ensure the robustness of our reported performance improvements. For each variant of CrysLDNet, we conduct five independent runs with different random seeds and report the mean, standard deviation, and 95\% confidence interval (CI). In addition, for each backbone, we compute paired t-test p-values to assess the statistical significance of the improvements. Specifically, we select PDDFormer, iComFormer, eComFormer, and Matformer, along with the corresponding variants of CrysLDNet that use these encoders as backbones, to evaluate statistical significance. The complete results on JARVIS dataset are as follows:
>
> | **Property** | **PDDFormer** | **CrysLDNet(PDDFormer)** |  |  | **iComFormer** | **CrysLDNet(iComFormer)** |  |  |
> |:-------------|:-------------:|:-----------------------:|:--:|:--:|:-------------:|:-----------------------:|:--:|:--:|
> |              |               | **Mean ± Std** | **CI** | **P-Value** |               | **Mean ± Std** | **CI** | **P-Value** |
> | Formation Energy | 0.027 | 0.026 ± 0.0004 | (0.0255, 0.0265) | 0.005 | 0.0272 | 0.0270 ± 0.0001 | (0.0269, 0.0271) | 0.011 |
> | Bandgap | 0.120 | 0.118 ± 0.001 | (0.117, 0.119) | 0.011 | 0.122 | 0.116 ± 0.002 | (0.114, 0.118) | 0.003 |
> | Total Energy | 0.028 | 0.027 ± 0.0002 | (0.0268, 0.0272) | 0.0003 | 0.029 | 0.028 ± 0.0003 | (0.0276, 0.0284) | 0.002 |
> | Ehull | 0.033 | 0.032 ± 0.0005 | (0.031, 0.033) | 0.011 | 0.044 | 0.036 ± 0.003 | (0.032, 0.040) | 0.004 |
> | mbj Bandgap | 0.251 | 0.242 ± 0.004 | (0.237, 0.247) | 0.007 | 0.261 | 0.240 ± 0.007 | (0.231, 0.249) | 0.003 |
> | Bulk Modulus | 9.546 | 8.817 ± 0.240 | (8.519, 9.115) | 0.0024 | 9.617 | 9.099 ± 0.200 | (8.851, 9.347) | 0.004 |
> | Shear Modulus | 8.808 | 8.528 ± 0.100 | (8.404, 8.652) | 0.003 | 9.098 | 8.966 ± 0.080 | (8.867, 9.065) | 0.020 |
> | SLME | 4.300 | 4.256 ± 0.010 | (4.244, 4.268) | 0.0006 | 4.583 | 4.529 ± 0.020 | (4.504, 4.554) | 0.004 |
> | Spillage | 0.358 | 0.340 ± 0.007 | (0.331, 0.349) | 0.004 | 0.360 | 0.340 ± 0.009 | (0.329, 0.351) | 0.007 |
>
>
> | **Property** | **eComFormer** | **CrysLDNet(eComFormer)** |  |  | **Matformer** | **CrysLDNet(Matformer)** |  |  |
> |:-------------|:---------------:|:-----------------------:|:--:|:--:|:--------------:|:----------------------:|:--:|:--:|
> |              |                 | **Mean ± Std** | **CI** | **P-Value** |                 | **Mean ± Std** | **CI** | **P-Value** |
> | Formation Energy | 0.0284 | 0.0280 ± 0.0003 | (0.0276, 0.0284) | 0.040 | 0.0325 | 0.029 ± 0.001 | (0.028, 0.030) | 0.001 |
> | Bandgap | 0.124 | 0.122 ± 0.001 | (0.121, 0.123) | 0.011 | 0.137 | 0.120 ± 0.010 | (0.108, 0.132) | 0.019 |
> | Total Energy | 0.0324 | 0.0320 ± 0.0003 | (0.0316, 0.0324) | 0.041 | 0.035 | 0.029 ± 0.002 | (0.027, 0.031) | 0.003 |
> | Ehull | 0.047 | 0.040 ± 0.004 | (0.035, 0.045) | 0.017 | 0.064 | 0.045 ± 0.009 | (0.034, 0.056) | 0.010 |
> | mbj Bandgap | 0.280 | 0.256 ± 0.009 | (0.245, 0.267) | 0.004 | 0.300 | 0.280 ± 0.010 | (0.268, 0.292) | 0.011 |
> | Bulk Modulus | 10.79 | 9.140 ± 1.200 | (7.650, 10.630) | 0.040 | 11.21 | 9.818 ± 1.000 | (8.576, 11.060) | 0.035 |
> | Shear Modulus | 9.826 | 9.422 ± 0.200 | (9.174, 9.670) | 0.011 | 10.76 | 9.108 ± 1.100 | (7.742, 10.474) | 0.028 |
> | SLME | 4.610 | 4.415 ± 0.090 | (4.303, 4.527) | 0.008 | 5.260 | 4.636 ± 0.400 | (4.139, 5.133) | 0.025 |
> | Spillage | 0.373 | 0.362 ± 0.004 | (0.357, 0.367) | 0.003 | 0.398 | 0.349 ± 0.010 | (0.337, 0.361) | 0.0004 |
>
> These statistical measures demonstrate that the performance gains achieved by CrysLDNet are consistent and reproducible across multiple runs. Notably, the paired t-tests yield p-values below 0.05 for most evaluated properties, confirming that the improvements are statistically significant. Overall, this analysis verifies that the superiority of CrysLDNet does not arise from random variation but reflects genuine and meaningful performance improvements across downstream tasks.
>
> We have updated these results in **Appendix  D.1.4 STATISTICAL SIGNIFICANCE OF THE RESULTS** in the revised manuscript.

---

### Official Review · Reviewer_zn3k · 2025-10-31

**Soundness:** 3
**Presentation:** 3
**Contribution:** 3
**Rating:** 6
**Confidence:** 4

**Summary:**

This paper proposes CrysLDNet, a Latent Diffusion pretraining framework for crystal property prediction. It addresses the limitations of existing diffusion models (like DPF) that operate directly on the non-smooth, high-dimensional input space. CrysLDNet is a two-stage process: 1) A VAE (with a Matformer encoder) compresses the crystal structure into a smooth latent space $Z$. 2) An LDM (DiT architecture) pretrains on this latent space $Z$ via denoising to optimize the encoder. The pretrained encoder $\mathcal{E}_{\phi}$ is then finetuned for downstream tasks

**Strengths:**

- Novel Framework: Moving diffusion from the high-D input space to a VAE latent space is well-motivated and addresses clear limitations of prior work .

- SOTA Performance: Comprehensively outperforms all baselines, including DPF and Matformer, on all 13 property tasks across JARVIS and MP (e.g., 6.93% & 7.83% average improvement).

- Data-Efficient: Demonstrates strong robustness and practical value in sparse-data regimes (Fig 4) and when correcting DFT errors with limited experimental data.

**Weaknesses:**

- Baseline Fairness: CrysLDNet uses the strong Matformer as its encoder. It is unclear if SOTA baselines (like DPF) use the same backbone, making the source of the performance gain (latent diffusion vs. better backbone) ambiguous.

- Pretraining Complexity: The two-stage (VAE + LDM) process is more complex than single-stage models like DPF; the associated training cost is not discussed.

- Unclear Ablation: The "LDM Only" ablation  is ambiguously described, making its distinction from the second stage of the full CrysLDNet pipeline unclear.

**Questions:**

- Baseline Encoder: What GNN encoder does the SOTA baseline DPF  use? How can you ensure the gains are from your latent diffusion framework and not just the Matformer backbone?

- "LDM Only" Ablation: Please clarify the "LDM Only" ablation setting. Did it start with a randomly initialized Matformer encoder?

---

> ### Author Response · Authors · 2025-11-22
> **Point-by-point responses to Reviewer zn3k**
>
> We thank the reviewer for the valuable feedback. Below, we provide detailed point-by-point responses to each of the questions:
>
> - ***Response to Concern on Baseline Fairness and Encoder Backbones.***
>
> > We thank the reviewer for the insightful comment. The baselines used in our evaluation do not share a common backbone. In the pretrain–finetune setting, our closest competitor DPF is built on Matformer, whereas other methods, such as CrysDiff use EGNN, while CrysXPP, CrystalTwins, and CrysGNN rely on simple GNNs. In the supervised (train-from-scratch) setting, only Matformer uses the same backbone; the remaining models rely on diverse GNN designs.
> >
> > Since the strongest and most relevant baselines in both categories (DPF and Matformer) use a Matformer encoder, we also adopt Matformer as our backbone to ensure a fair and consistent comparison.
> >
> > We emphasize that our method is backbone-agnostic, and we further observe consistent performance gains when using more recent encoders such as iComformer or PDDformer. These additional baseline results have been included in the revised manuscript.
>
> - ***Regarding Pretraining Complexities.***
>
> > We agree with the reviewer that our two-stage pretraining (VAE + LDM) is somewhat more expensive than a single-stage model like DPF. For clarity, we report the total  pre-training time of DPF and Our framework on an L40 GPU server:
> >
> > `DPF requires 377 minutes in total, while our VAE and LDM stages in CrysLDNet take 210 minutes and 301 minutes respectively, for a total of 511 minutes. `
> >
> > Overall, this cost remains manageable in practice and is only incurred once during pretraining. We have added a brief discussion of this in the revised manuscript.
>
> - ***Regarding Unclear Ablation "LDM Only".***
>
> > We thank the reviewer for raising this question, and we confirm their observation: in the “LDM Only” ablation setting, we indeed begin with a randomly initialized Matformer encoder.
> The key difference between the second stage of the full CrysLDNet pipeline and the LDM-Only baseline lies in the state of the encoder feeding the latent diffusion model. In the full CrysLDNet pipeline, the VAE is first pretrained using the reconstruction loss (Eq. 3), and the resulting encoder is then further refined during the latent diffusion training. Thus, the LDM operates on a pretrained and semantically meaningful latent space.
> >
> > In contrast, the LDM-Only model does not include a VAE. Instead, it uses a MatFormer encoder with randomly initialized parameters to produce latent representations, and the latent diffusion model operated on that. Both the LDM and MatFormer Encoder is trained jointly from scratch. As a result, the encoder in this setting lacks the structured inductive biases learned during VAE pretraining, which explains the observed performance gap between the full CrysLDNet pipeline and the LDM-Only configuration.
> A brief discussion reflecting this clarification has also been added to the updated version.

---

### Official Review · Reviewer_9Mva · 2025-11-01

**Soundness:** 3
**Presentation:** 3
**Contribution:** 3
**Rating:** 6
**Confidence:** 3

**Summary:**

This paper proposes a new pretraining/finetuning method for crystal property prediction. The goal of pretraining is obtaining a good feature encoder of crystal structures from an unlabeled dataset; for this the authors first train a standard VAE, and then further train the encoder along with a flow network using conditional flow matching loss based on optimal transport conditional vector field. At fine-tuning stage, the encoder is postfixed with a property prediction head and fine-tuned with a labeled dataset. On the standard benchmark setups including both synthetic and experimental data, as well as in scarce data regimes, the authors report an improved performance over state-of-the-art baselines.

**Strengths:**

S1. The proposed method is, while being a combination of existing components, technically sound as far as I can confirm (with one caveat in W1), and it empirically achieves strong performances in various settings compared to existing methods. The empirical evaluation setup is sound as far as I understand; the baselines include recent state-of-the-art methods, and the hyperparameters for fine-tuning are chosen consistently across datasets (Appendix C.1).

**Weaknesses:**

W1. In Section 4.2.2, the authors claim that the VAE encoder and the flow network (learned vector field) are trained jointly with the matching loss in Equation (5). Usually, this type of training can easily collapse due to the presence a local optima, where the VAE encoder can simply collapse to produce a constant output, because then it becomes trivial for the flow network to reach zero loss by predicting the direction towards the constant target. Proper discussion and/or analysis on why such phenomenon does not happen seem necessary.

W2. Can the authors discuss why/how latent flow matching improves the quality of representation from the VAE encoder? This is validated in Table 3 but its underlying reason is not immediate to me.

W3. This is a minor comment, but I believe it is necessary to cite the original flow matching paper because the noise schedule and regression objective function are borrowed from there. Although OT flow matching is dual to a Gaussian diffusion, regressing the vector field is essentially taking the flow matching and probability flow ODE viewpoint.

**Questions:**

I don't have particular questions but would like to hear the authors' opinion on the weaknesses.

---

> ### Author Response · Authors · 2025-11-22
> **Point-by-point responses to Reviewer 9Mva (1/2)**
>
> We thank the reviewer for the valuable feedback. Below, we provide detailed point-by-point responses to each of the questions:
>
> - ***Response to Comment on Joint VAE–Flow Training Stability.***
>
> > We thank the reviewer for raising this concern. We agree that joint VAE–Flow training can become unstable and may cause the encoder to collapse, but this usually happens when the encoder is trained from scratch with random initialization [1][2]. A common and effective solution is to pretrain the encoder so that it starts from a meaningful state. In our case, the encoder is not trained from scratch, rather we first pretrain the encoder using a VAE with a reconstruction loss over atom types, coordinates, and lattice, along with a KL regularizer (Eq. 3). Only after this step, the encoder is further refined during joint training with the LDM (Algorithm 1, Stage 2). This warm start with VAE ensures that the encoder already produces meaningful, non-collapsed latent representations, which the LDM then improves rather than driving toward a trivial constant output.
> >
> > However, to directly test the reviewer’s concern, we compared the losses of (i) a standard LDM trained without our pretrained encoder(without VAE) and (ii) our full CrysLDNet model, and here are the results for 20 epochs
> >
> >| Epoch | LDM (without VAE) Loss | CrysLDNet Loss |
> >|-------|--------------------------|------------------|
> >| 1  | 1.43440 | 5.39335 |
> >| 2  | 0.16141 | 0.88336 |
> >| 3  | 0.07241 | 0.32532 |
> >| 4  | 0.03415 | 0.11885 |
> >| 5  | 0.01984 | 0.06395 |
> >| 6  | 0.01455 | 0.02581 |
> >| 7  | 0.00902 | 0.02445 |
> >| 8  | 0.00601 | 0.02373 |
> >| 9  | 0.00505 | 0.00834 |
> >| 10 | 0.00407 | 0.01135 |
> >| 11 | 0.00472 | 0.00536 |
> >| 12 | 0.00149 | 0.00395 |
> >| 13 | 0.00912 | 0.00430 |
> >| 14 | 0.00156 | 0.00270 |
> >| 15 | 0.00212 | 0.00447 |
> >| 16 | 0.00192 | 0.00248 |
> >| 17 | 0.00094 | 0.00261 |
> >| 18 | 0.00094 | 0.00189 |
> >| 19 | 0.00075 | 0.00125 |
> >| 20 | 0.00063 | 0.00320 |
> >
> > In the first setting, the loss collapses almost immediately: it drops from 1.43 → 0.16 → 0.07 → 0.03 within just the first 4 epochs, and reaches the order of 10⁻³–10⁻⁴ by epoch 12 (e.g., 0.00148 at epoch 12 and 0.00063 at epoch 20). By epoch 50, the loss is as low as 1.7×10⁻⁴, indicating a near-trivial solution. This extremely rapid convergence and vanishing loss are consistent with the encoder collapsing to an almost constant latent, allowing the flow network to minimize the matching loss trivially.
> >
> >In sharp contrast, CrysLDNet does not exhibit this behavior. Its loss decreases much more gradually: from 5.39 (epoch 1) to 0.88, 0.32, and 0.11 across the first four epochs, and stabilizes in the range of 10⁻³–10⁻² during mid-training (e.g., 0.0083 at epoch 9, 0.0039 at epoch 12, 0.0026 at epoch 17). Even at epoch 20, the loss remains 0.00319, significantly larger than the collapsed LDM baseline (0.00063).
> >
> >We have provided the details of the losses in Appendix D (Joint VAE–Flow Training Stability) of the revised manuscript. This comparison clearly shows that collapse occurs only when the encoder lacks reconstruction/KL constraints (LDM without VAE), whereas the full CrysLDNet training remains stable and avoids the degenerate constant-latent solution the reviewer described.
> >
> >
> >[1] Chien, Jen-Tzung, and Tien-Ching Luo. "Flow-Based Variational Sequence Autoencoder." 2022 Asia-Pacific Signal and Information Processing Association Annual Summit and Conference (APSIPA ASC). IEEE, 2022.
> >
> >
> >[2] Bhattacharyya, Apratim, et al. "Conditional flow variational autoencoders for structured sequence prediction." arXiv preprint arXiv:1908.09008 (2019).

---

> ### Author Response · Authors · 2025-11-22
> **Point-by-point responses to Reviewer 9Mva (2/2)**
>
> - ***Response to Concern on How/Why Latent Flow Matching Improves Encoder Representations.***
> > **Mutual Information Analysis Between Learned Representations and Material Structure:** We measured the Mutual Information (MI) between the learned latent representations and the underlying material structure to quantify how much structural information the encoder retains [1], comparing the VAE-only model with CrysLDNet (VAE+LDM). CrysLDNet exhibits substantially higher MI with both atom types (VAE: 3.0906 $\rightarrow$ CrysLDNet: 4.5465) and atomic coordinates (VAE: 1.3124 $\rightarrow$ CrysLDNet: 2.4864), indicating that the LDM refinement produces richer and more structurally grounded embeddings..
> >
> > [1] Hjelm, R. Devon, et al. "Learning deep representations by mutual information estimation and maximization." arXiv preprint arXiv:1808.06670 (2018).
> >
> > **Ablation Studies on VAE Latents Across Epochs:** We posit that more expressive latent representations directly translate into improved downstream property-prediction performance. In our framework, the VAE provides an initial latent encoding, which is subsequently refined by the latent diffusion model (LDM), yielding richer and more informative representations. This is consistent with the results in Table 3, where CrysLDNet outperforms the VAE-only baseline.
> To examine this effect in greater detail, we performed an ablation study in which VAE latents were progressively refined by the LDM for varying numbers of training epochs. We observed that, with increasing epochs, the LDM consistently enhanced the VAE representations, leading to monotonic improvements in property-prediction accuracy. Below, we report representative results on several properties from the JARVIS dataset:
> >
> >| Epoch | Bulk Modulus | Shear Modulus | SLME (%) | Spillage |
> >|-------|--------------|---------------|----------|----------|
> >| 1     | 10.60       | 9.767         | 4.818    | 0.367    |
> >| 3     | 10.41       | 9.700         | 4.781    | 0.362    |
> >| 5     | 10.35       | 9.674         | 4.740    | 0.356    |
> >| 10    | 10.27       | 9.655         | 4.695    | 0.355    |
> >| 20    | 10.16       | 9.492         | 4.651    | 0.352    |
> >| 30    | 9.912        | 9.369         | 4.647    | 0.351    |
> >| 50    | **9.818**        |**9.108**         |**4.636**   | **0.349**    |
> >
>
>
> - ***Response to Citation for Original Flow Matching Work.***
> > We thank the reviewer for pointing this out, and we have added a citation to the original Flow Matching work [2] in the revised manuscript.
> >
> > [2] Lipman, Y., Chen, R. T., Ben-Hamu, H., Nickel, M., & Le, M. (2022). Flow matching for generative modeling. arXiv preprint arXiv:2210.02747.

---

> ### Comment · Reviewer_9Mva · 2025-11-25
>
> Thank you for the thorough response, which has resolved W1 and W3.
>
> On W2 (why/how latent flow matching improves the quality of representation), to be clear, am I correct in understanding that LDM is used to fine-tune the VAE encoder, and then is discarded for property prediction fine-tuning, where only the (refined) VAE encoder is used? The two added experiments strengthen the results of ablation studies that this LDM fine-tuning of the VAE encoder improves the quality of representations ("richer and more informative"), but the explanation provided are not entirely satisfactory; why should LDM fine-tuning lead to improvements? The authors claim "more expressive latent representations directly translate into improved downstream property-prediction performance", but if LDM is discarded and only the VAE encoder is used for property prediction, there is no effective gain in expressive power of the neural network architecture used, so I don't see how this explanation is correct.
>
> Minor comments
> - Lines 394-395: all of which are yield -> all of which yield
> - Table 6 would be more readable if it was a plot.

---

> > ### Author Response · Authors · 2025-11-26
> >
> > We thank the reviewer for the positive feedback.
> >
> > The purpose of using an LDM is not to increase the model’s expressiveness, but to refine the latent geometry learned by the encoder. Our objective is to pretrain the encoder so that it captures meaningful structural and chemical semantics from a large corpus of unlabeled materials data. To achieve this, we adopt a two-stage pretraining strategy inspired by latent diffusion models.
> >
> > ***Stage 1 (VAE Pretraining):***
> >
> >  We begin with a VAE framework by pairing the encoder with a decoder and training them using a reconstruction objective. This stage encourages the encoder to learn fundamental semantics of the input structures. Once this pretraining is complete, the decoder has fulfilled its role and is discarded; we retain only the encoder.
> >
> > ***Stage 2 (Latent Diffusion refinement):***
> >
> >  Next, we take the pretrained encoder and apply a latent diffusion model on its embeddings. Specifically, a transformer-based denoising network is trained using a denoising objective, and through an end-to-end joint training of VAE-encoder and LDM,  enabling the encoder to better align its latent representations with the underlying data distribution in a smooth latent space. After this refinement, the denoising network is also discarded.
> >
> > In both stages, the auxiliary components (decoder and denoiser) serve only to guide the encoder toward richer, more informative representations. The final encoder—now enhanced with deeper structural and chemical signals—is used for downstream property prediction.  We have included a simplified block diagram illustrating the overall CrysLDNet pipeline in Figure 6 of the Appendix.
> >
> > All minor comments have also been addressed in the revised manuscript. Specifically, Table 6 has been replaced with Figure 5 in the Appendix.

---

> > > ### Comment · Reviewer_9Mva · 2025-11-26
> > >
> > > Thank you for the follow-up and incorporating the suggestions. After reading the responses and other reviews, my assessment is that this is a technically solid work presenting a simple and architecture-agnostic method that empirically works consistently well (my assessment on performances might be limited, as I am more from the architecture research domain rather than chemistry). Hence, I would like to retain my supportive rating.
> > >
> > > My reason for not assigning a higher score is that W2 remains somewhat unclear to me; the empirical results support that latent diffusion is helpful in some way but the underlying reason or mechanism is not clearly revealed in the paper, even with the authors' added explanations, that I find somewhat vague (to be more specific and constructive, I wasn't able to clearly grasp: what does "richer representations" or "better aligned representations with the underlying data distributions" specifically mean? and why can't the VAE learn them directly, and why is diffusion needed to learn such representations?).

---

> ### Author Response · Authors · 2025-12-03
>
> ***Why Diffusion Helps: Clarifying “Richer” and “Better-Aligned” Representations***
>
> A richer representation is one that captures deeper semantic and structural information, such as composition–structure relationships, geometric nuances, and other patterns that are useful for downstream tasks like prediction, generation, or editing. In other words, a representation is better aligned with the underlying crystal data distribution when it reflects the true complexity, diversity, and multimodal nature of crystal structures.
>
> We are not claiming that a VAE cannot learn such representations. In fact, our VAE-only model (Table 3) does learn meaningful structure and performs reasonably well. However, the crystal space is highly multimodal, spanning diverse space groups, symmetries, and lattice families. A standalone VAE has modest ability to capture these global structural variations and complex invariances in 3D crystal geometry.
>
> By introducing a diffusion model on the latent space, we enable the model to better learn these multimodal regions and high-level semantics [1]. Empirically, our ablation results in Table 3 show that CrysLDNet (VAE + LDM) consistently outperforms the VAE-only baseline across all evaluated properties, demonstrating the advantage of combining diffusion with a VAE. We believe these added explanatory insights will clarify the reviewer's doubt.
>
> [1] Liu, Guangyi, et al. "Unified generation, reconstruction, and representation: generalized diffusion with adaptive latent encoding-decoding." ICML - 2024.

---

### Official Review · Reviewer_TTE4 · 2025-11-01

**Soundness:** 3
**Presentation:** 3
**Contribution:** 3
**Rating:** 8
**Confidence:** 4

**Summary:**

The authors propose CrysLDNet, a novel pretrain-finetune framework for crystal property prediction designed to address the problem of labeled-data scarcity . The core idea is to move the pretraining task from the high-dimensional, heterogeneous input space (atom types, coordinates, lattice) to a more compact and "smooth" latent space.

**Strengths:**

1. It is novel to consider the latent diffusion as a pre-training task for a downstream property predictor. The paper correctly identifies a key weakness in existing diffusion pretraining models like DPF.

2. CrysLDNet achieves state-of-the-art results on all 13 property prediction tasks across both the JARVIS and Materials Project datasets

3. The paper is very well-written. The introduction clearly motivates the problem, the related work section precisely identifies the limitations of prior art, and the proposed method (Figure 2) is logical and easy to follow.

**Weaknesses:**

The entire focus of the paper is based on the claim that VAE creates a "smooth" and "compact" latent space $Z$, which is then "refined" and "enriched" by the LDM. It is unclear about the definition of smoothness or compactness. Without this in-depth analysis, the success of this model is still a black box.

**Questions:**

1. Can you please provide an analysis of the latent space learned by the encoder?

2. You claim the LDM "refines" and "enriches" the latent representations. Following up on Question 1, can you provide any evidence of this?

3. Should line 16 in Algorithm 1 be $Z^0 \sim \mathcal{N}(0, I)^{N \times d}$ to match the per-node loss in line 19?

---

> ### Author Response · Authors · 2025-11-22
> **Point-by-point responses to Reviewer TTE4**
>
> We thank the reviewer for the valuable feedback. Below, we provide detailed point-by-point responses to each of the questions:
> - ***line 16 in Algorithm 1***
> > We thank the reviewer for pointing out. We have changed in the revised manuscript.
>
> - ***Analysis of the latent space learned by the Encoder.***
> > **Mutual Information Analysis Between Learned Representations and Material Structure:** We measured the Mutual Information (MI) between the learned latent representations and the underlying material structure to quantify how much structural information the encoder retains [1], comparing the VAE-only model with CrysLDNet (VAE+LDM). CrysLDNet exhibits substantially higher MI with both atom types (VAE: 3.0906 $\rightarrow$ CrysLDNet: 4.5465) and atomic coordinates (VAE: 1.3124 $\rightarrow$ CrysLDNet: 2.4864), indicating that the LDM refinement produces richer and more structurally grounded embeddings..
> >
> > [1] Hjelm, R. Devon, et al. "Learning deep representations by mutual information estimation and maximization." arXiv preprint arXiv:1808.06670 (2018).
> >
> > **Ablation Studies on VAE Latents Across Epochs:** We posit that more expressive latent representations directly translate into improved downstream property-prediction performance. In our framework, the VAE provides an initial latent encoding, which is subsequently refined by the latent diffusion model (LDM), yielding richer and more informative representations. This is consistent with the results in Table 3, where CrysLDNet outperforms the VAE-only baseline.
> To examine this effect in greater detail, we performed an ablation study in which VAE latents were progressively refined by the LDM for varying numbers of training epochs. We observed that, with increasing epochs, the LDM consistently enhanced the VAE representations, leading to monotonic improvements in property-prediction accuracy. Below, we report representative results on several properties from the JARVIS dataset:
> >
> >| Epoch | Bulk Modulus | Shear Modulus | SLME (%) | Spillage |
> >|-------|--------------|---------------|----------|----------|
> >| 1     | 10.60       | 9.767         | 4.818    | 0.367    |
> >| 3     | 10.41       | 9.700         | 4.781    | 0.362    |
> >| 5     | 10.35       | 9.674         | 4.740    | 0.356    |
> >| 10    | 10.27       | 9.655         | 4.695    | 0.355    |
> >| 20    | 10.16       | 9.492         | 4.650    | 0.352    |
> >| 30    | 9.912        | 9.369         | 4.647    | 0.351    |
> >| 50    | **9.818**        | **9.108**         | **4.636**    | **0.349**    |
> >
>
> - ***Evidence of LDM refines and enriches the latent representations.***
> > As reported in Table 3, we include ablation studies comparing the VAE-only model with the full CrysLDNet. In the VAE-only variant, we employ only the VAE component without any LDM refinement. Across all evaluated properties on the JARVIS dataset (in Table 3), CrysLDNet consistently outperforms the VAE-only model. This clearly demonstrates that the VAE latent representations are further refined and enriched through the LDM refinement step in CrysLDNet.

---

> > ### Comment · Reviewer_TTE4 · 2025-11-26
> >
> > Thanks for the response. I maintain the score after checking the rebuttal and reviews from other reviewers.

---

### Author Response · Authors · 2025-11-22
**Summary of Revised Manuscript**

We sincerely thank the reviewers for their constructive and insightful feedback. In response, we have thoroughly revised both the main manuscript and the appendix, with **all modifications highlighted in blue**. A summary of the key updates is provided below:

**Key Revisions**

- ***Expanded Evaluation with Additional Baselines:*** As suggested by the reviewers, we have incorporated several newer and stronger baseline models into Table 2 and added a corresponding discussion in the section “Results on New Baselines” (Line 399).


- ***Backbone-Agnostic Design Clarification:*** Addressing Reviewer Akui’s comment, we have added a dedicated paragraph titled “Backbone-Agnostic Design” (Lines 287–295), explicitly stating that CrysLDNet is fully agnostic to the choice of encoder backbone. We also updated Figure 2 to clearly highlight the encoder as a modular, interchangeable component, and revised the caption to emphasize this design choice.


- ***Comprehensive Statistical Analysis:*** In response to Reviewer Akui, we added a new section D.1.4 STATISTICAL SIGNIFICANCE OF THE RESULTS in the appendix, along with Table 8, providing a detailed statistical significance analysis for all reported results.


- ***Enhanced Latent Representation Analysis:*** To address concerns regarding reconstruction accuracy in Fig. 3, we updated the figure to include results from both the Materials Genome and JARVIS datasets. The corresponding discussion is now provided in “Expressiveness of Latent Representations” (Line 420/21).


- ***Clarified “LDM Only” Ablation:*** In Section 5.3 ABLATION STUDY, we have added a clear and concise explanation of the “LDM Only” ablation to resolve the reviewer’s confusion (Lines 481–495).


- ***New Latent Space Analysis Section:*** We introduced a new subsection, 5.4 ANALYSIS OF THE LATENT SPACE LEARNED BY THE ENCODER (Line 507), offering deeper insights into the structure and behavior of the learned latent representations.


- ***Pretraining Complexity Comparison:*** Section 6. PRETRAINING COMPLEXITIES (Line 522) now includes a detailed comparison between the computational cost of our pretraining pipeline and that of the diffusion-based pretrained model DPF. Additional details are provided in Appendix Table 8.


- ***Discussion on Joint VAE–Flow Training Stability:*** We added a new analysis in Appendix C:  JOINT VAE–FLOW TRAINING STABILITY, discussing the instabilities encountered during joint VAE–Flow training, along with the stabilization strategy adopted in our framework.


- ***Relocation of Experimental Data Results:*** To meet page-limit requirements, the section “RESULTS USING EXPERIMENTAL DATA” has been moved to Appendix D.1.3.

- We have included a simplified diagram illustrating the overall CrysLDNet pipeline in Figure 6 of the Appendix.


These revisions address all reviewer concerns and considerably strengthen the manuscript in terms of clarity, completeness, and robustness.

---

### Author Response · Authors · 2025-12-03
**Summary of the Reviewer Discussions**

Dear ACs and Senior ACs,

We are pleased to inform you that we got some insightful feedback from all the reviewers, Almost all reviewers appreciated the work  and found the work to be relevant. We have answered all their concerns and incorporated all necessary changes in the revised manuscript. Below is a summary of our discussions with each reviewer.

# Reviewer TTE4 :
The reviewer acknowledged several strengths of our work, including the novelty of using latent diffusion as a pre-training objective. The reviewer requested clarifications and a few new results as detailed below.

- **New Results Asked:** An analysis of the encoder’s latent representations and evidence supporting our claim that the latent diffusion model further “refines” or “enriches” these embeddings.
>
>Results are Provided. (Line-507-519, in Section 5.4 in Revised Manuscript)
>
>Reviewer Acknowledged.

- **Clarification Asked:**  Clarification regarding a potential inconsistency in line 16 of Algorithm 1.
>
> Clarification is provided. (Line-224, Algorithm 1, in Revised Manuscript)
>
> Reviewer Acknowledged.

# Reviewer 9Mva:
The reviewer highlighted that our proposed method is technically sound and demonstrates strong empirical performance across datasets. The reviewer requested a few new results, clarifications, and minor revisions, as detailed below.

- **Clarification Asked:**   Potential collapse when jointly training the VAE encoder and flow network with the matching loss.
>
>Clarification is provided.(in Appendix C in Revised Manuscript)
>
>Reviewer Acknowledged

- **Suggested Minor Revisions:**  Citing the original flow-matching paper.
>
>Cited the paper in the revised manuscript.
>
>Reviewer Acknowledged

- **New Results Asked:** Deeper explanation/insights of why and how latent flow matching improves the VAE encoder’s representations.
>
> New results provided (Line-507-519, in Section 5.4 in Revised Manuscript).
>
> Reviewer Acknowledged.
>
> Further explanatory insights are provided.

# Reviewer zn3k:
The reviewer noted that shifting diffusion from the high-dimensional input space to a VAE latent space directly addresses limitations of prior approaches. The reviewer requested new results and clarifications as detailed below.

- **Clarification Asked:** Regarding the fairness of baseline comparisons.
>
> Clarifications are provided.

- **New Results Asked:** Complexity of pretraining pipeline of CrysLDNet.
>
>New results and further clarifications are provided. (Line-522-528, in Section 6 in Revised Manuscript)

- **Clarification Asked:** Ambiguity in the description of the “LDM Only” ablation.
>
> Clarifications are provided (Line-481-495 in Revised Manuscript).

# Reviewer Akui:
The Reviewer highlighted several strengths of our submission—including the model’s simple and interpretable design, consistent improvements over baselines, and its relevance to challenges of data sparsity and generalization in materials informatics.

### Round-1

- **Clarification Asked:** Clarification regarding VAE formulation.
>
>Clarification is provided.
>
>Reviewer Acknowledged

- **Clarification Asked:** Specific denoising or diffusion steps are used in the latent pretraining.
>
>Clarification is provided.
>
>Reviewer Acknowledged

- **New results and Clarification Asked:** Justification regarding reconstruction quality and preserving important physical and structural information.
>
>New results and further clarifications are provided (Line-421-459 in Revised Manuscript).
>
>Reviewer Acknowledged

- **New Results Asked:** New baselines for empirical evaluation.
>
>New results are provided (Line 398-419 in Revised Manuscript).
>
>Reviewer Acknowledged

### Round-2

- **Recommended revision:** Adding clarifying content on backbone-agnostic Design of CrysLDNet.
>
> Addition Done (Line 287-295 in Revised Manuscript, changes in Fig-2)

- **New Results Asked:** Statistical Analysis of the newly added baseline results.
>
> Statistical analyses are provided (In Appendix D.1.4 in Revised Manuscript).

- **New Results Asked:** Computational cost of CrysLDNet
>
> New results and further clarifications are provided (Line 524-528 in Revised Manuscript).

- The reviewer also noted the absence of certain baselines in the original submission. We have now included these baselines during the rebuttal phase, and they do not alter the main conclusions of the paper. We regard the rebuttal period, enabled by ICLR’s policy, as an appropriate opportunity to incorporate such additions and further clarify our results.

We are encouraged by the outcome of the discussion phase, and we have addressed all the concerns raised by the reviewers. The constructive nature of the exchanges has helped us further strengthen our work, both in terms of technical rigor and presentation. We believe the revised manuscript will reflect these improvements and make a strong contribution to the field.

Thank you,

The Authors

---

### Meta-Review · Area_Chair_xfbq · 2026-01-06

**Summary:**

This paper introduces CrysLDNet, a novel pretraining framework for crystal property prediction that addresses data scarcity. Unlike existing methods that apply diffusion in the heterogeneous input space, CrysLDNet employs a two-stage approach: a VAE first maps structures to a smooth latent space, followed by a Latent Diffusion Model (LDM) to refine these representations. This strategy captures richer structural semantics, achieving state-of-the-art performance on JARVIS and Materials Project benchmarks.

Strengths:

(1) Introduces the first latent diffusion-based pretraining framework for crystal structure prediction.
(2) Achieves significant performance gains over existing baselines on JARVIS and Materials Project benchmarks.

Weaknesses of the initial version:
(1) Initial evaluations lacked comparisons against the most recent state-of-the-art transformer baselines.
(2) There was no analysis of the training stability of the joint VAE–Diffusion framework, nor any discussion of the potential risk of encoder collapse.
(3) The work did not include experiments to assess whether performing diffusion in the latent space leads to superior structural reconstruction compared to applying diffusion directly in the data space.

**Reviewer Concerns:**

The authors have provided a robust rebuttal that effectively addresses several key concerns:

(1) They demonstrated the "backbone-agnostic" nature of their method, showing consistent gains against improved baselines (e.g., PDDformer, eComformer).

(2) To address concerns about mode collapse in joint VAE-Diffusion training, they provided loss comparison curves (Figure 5), proving that their two-stage "warm start" strategy ensures stability where single-stage methods might fail.

(3) New experiments (Figure 3) verified that latent-space diffusion yields superior reconstruction capabilities compared to feature-space approaches like CrysDiff and DPF.

However, two central concerns remain outstanding.

(1) The core methodological contribution remains the combination of a VAE and Latent Diffusion for pre-training a crystal encoder. This integration, while competently executed, does not constitute a significant theoretical or architectural advance within the field.

(2) Given the substantial computational cost of the proposed pre-training process, the overall performance improvement over a train-from-scratch baseline using the same backbone is marginal. The addition of an MLP to the encoder inherently increases model capacity; therefore, some performance gain is expected simply from the enlarged parameter size, making the added value of the complex pre-training pipeline unclear.

In addition, the authors state they keep "all other hyperparameters identical to those used during pre-training." This is questionable, as different datasets typically require different training dynamics. The rationale for using an identical hyperparameter set across distinct datasets and tasks is not justified.

**Reviewer Scores:**

The initial review scores were: TTE4 (8), 9Mva (6), zn3k (6), and Akui (2).  While the authors' rebuttal has addressed some concerns from the reviewer who scored a 2, a potential score increase to a 4 would not alter the paper's overall standing. Given the persisting major concerns about technical novelty and experimental significance, the consensus assessment remains negative.

---

### Decision · Program_Chairs · 2026-01-26

Reject